# Mechanism of action and therapeutic route for a muscular dystrophy caused by a genetic defect in lipid metabolism

Mahtab Tavasoli [1], Sarah Lahire [2], Stanislav Sokolenko[3], Robyn Novorolsky[1], Sarah Anne Reid[1], Abir Lefsay[4], Meredith O. C. Otley[1], Kitipong Uaesoontrachoon[5], Joyce Rowsell[5], Sadish Srinivassane[5], Molly Praest[5], Alexandra MacKinnon[5], Melissa Stella Mammoliti[5], Ashley Alyssa Maloney[5], Marina Moraca[5], J. Pedro Fernandez-Murray[1], Meagan McKenna[5], Christopher J. Sinal[1], Kanneboyina Nagaraju[5,6], George S. Robertson[1,7], Eric P. Hoffman[5,6] & Christopher R. McMaster [1]✉

*CHKB* encodes one of two mammalian choline kinase enzymes that catalyze the first step in the synthesis of the membrane phospholipid phosphatidylcholine. In humans and mice, inactivation of the *CHKB* gene (*Chkb* in mice) causes a recessive rostral-to-caudal muscular dystrophy. Using *Chkb* knockout mice, we reveal that at no stage of the disease is phosphatidylcholine level significantly altered. We observe that in affected muscle a temporal change in lipid metabolism occurs with an initial inability to utilize fatty acids for energy via mitochondrial β-oxidation resulting in shunting of fatty acids into triacyglycerol as the disease progresses. There is a decrease in peroxisome proliferator-activated receptors and target gene expression specific to *Chkb*$^{-/-}$ affected muscle. Treatment of *Chkb*$^{-/-}$ myocytes with peroxisome proliferator-activated receptor agonists enables fatty acids to be used for β-oxidation and prevents triacyglyerol accumulation, while simultaneously increasing expression of the compensatory choline kinase alpha (*Chka*) isoform, preventing muscle cell injury.

[1] Department of Pharmacology, Dalhousie University, Halifax, NS, Canada. [2] University of Reims Champagne-Ardenne, Reims, France. [3] Department of Process Engineering & Applied Science, Dalhousie University, Halifax, NS, Canada. [4] Mass Spectrometry Core Facility, Dalhousie University, Halifax, NS, Canada. [5] Agada Biosciences Inc., Halifax, NS, Canada. [6] School of Pharmacy and Pharmaceutical Sciences, Binghamton University, State University of New York (SUNY), Binghamton, NY, USA. [7] Department of Psychiatry, Dalhousie University, Halifax, NS, Canada. ✉email: christopher.mcmaster@dal.ca

Phosphatidylcholine (PC) is the major phospholipid present in mammalian cells, comprising approximately 50% of phospholipid mass. Choline kinase catalyzes the phosphorylation of choline to phosphocholine and is the first enzymatic step in the synthesis of PC[1]. There are two genes that encode human choline kinase enzymes, *CHKA* and *CHKB*. Monomeric choline kinase proteins combine to form homo- or hetero-dimeric active forms[2]. CHKA and CHKB proteins share similar structures and enzyme activity but display some distinct molecular structural domains and differential tissue expression patterns. Knock-out of the murine *Chka* gene leads to embryonic lethality[3]. *Chkb* deficient (*Chkb$^{-/-}$*) mice are viable, but noticeably smaller than their wild-type counterparts, and show severe bowing of the ulna and radius at birth. By 2–3 months of age *Chkb$^{-/-}$* mice lose hindlimb motor control, while the forelimbs are spared[4,5]. Inactivation of the *Chkb* gene in mice would be predicted to decrease PC level, however, reports indicate no, or a very modest, decrease in PC level in *Chkb$^{-/-}$* mice, and this decrease is similar in both forelimb and hindlimb muscle[6,7]. The very small decrease in PC mass, and the fact that there is no rostral-to-caudal change in PC, suggest a poor correlation of the anticipated biochemical defects and observed rostral-to-caudal phenotype of this muscular dystrophy[5]. It is unclear how a defect in a gene required for the synthesis of a major phospholipid in mammalian cells causes a muscular dystrophy, especially given that global inactivation of the *CHKB/Chkb* gene (human or mouse) does not affect the level of the product of its biochemical pathway, PC.

Muscular dystrophy, congenital, megaconial type (OMIM 602541) is an autosomal recessive dystrophy caused by loss of function of *CHKB* gene and is the only defect in phospholipid synthesis that can cause a muscular dystrophy[5,8–14]. Muscular dystrophies have been mapped to at least 30 different causal genes[15]. The most common types of muscular dystrophy result from mutations in genes coding for members of protein complexes which act as linkers between the cytoskeleton of the muscle cell and the extracellular matrix that provides mechanical support to the plasma membrane during myofiber contraction[16,17]. Muscular dystrophies result in fibrofatty replacement of muscle tissue, progressive muscle weakness, functional disability, and often early death[18,19,20].

Skeletal muscle accounts for 20–30% of whole body basal metabolic rate[21]. Fatty acid oxidation is the major source of ATP for skeletal muscle during the resting state[22]. Fatty acids can be synthesized de novo by cells or can be obtained extracellularly, with the bulk of lipids delivered to cells through the circulation via serum albumin or lipoprotein receptors. For fatty acids to be metabolized they are first activated by esterification to fatty acyl-CoA. Subsequently, they have divergent fates depending on the metabolic status of the cells. The three major fates of fatty acids are (1) conversion to fatty acyl carnitine for subsequent mitochondrial β-oxidation to provide energy, (2) the synthesis of neutral lipid species for storage as triacylglycerol (TG) rich cytoplasmic lipid droplets, and (3) metabolism into phospholipids, such as PC, to maintain membrane integrity. Fatty acids can also directly bind peroxisome proliferator-activated receptors (Ppars), key players that regulate lipid metabolism by altering the expression of genes required for the conversion of fatty acids to fatty acyl-CoA for phospholipid and TG synthesis, and for fatty acid activation to acylcarnitine (AcCa) for entry into mitochondria and subsequent fatty acid β-oxidation[23].

In the present study, we used cell-based and mouse models to investigate the temporal changes in lipid metabolism in the absence of the *Chkb* gene. Our results demonstrate that *Chkb* deficiency did not alter PC levels. Instead, this genetic defect in PC synthesis produces large fluctuations in mitochondrial lipid metabolism and inability to use fatty acids for mitochondrial β-oxidation leading to a temporal shunting of fatty acids into TG and their storage as lipid droplets. These findings provide insights into the surprising biochemical phenotype whereby a genetic block in a phospholipid metabolic pathway does not directly affect the product of its pathway, and instead alters tangential pathways in a manner that explains the rostral-to-caudal gradient of a genetic disease. The unexpected alterations in this lipid metabolic profile suggest that Ppars may have a major role in metabolic disease etiology. We go on to demonstrate that Ppar agonists both reverse the defect in mitochondrial fatty acid β-oxidation and prevent the accumulation of lipid droplets caused by *Chkb* ablation. Interestingly, Ppar agonists also increase the expression of *Chka* enabling reconstitution of the CDP-choline pathway for the synthesis of PC. Lastly, the addition of choline and a Ppar agonist reduces injury to *Chkb$^{-/-}$* myocytes.

## Results

**Choline kinase-deficient mice display hallmark muscular dystrophy phenotypes.** To address the extent that mice lacking Chkb function display gross muscular dystrophy phenotypes, we assessed total lean and fat mass, as well as muscle function in *Chkb$^{+/+}$*, *Chkb$^{+/-}$* and *Chkb$^{-/-}$* mice from 6 weeks to 20 weeks of age using a grip strength assay and a total distance run to exhaustion test. Body weight was also recorded each week at similar times over the duration of the phenotyping experiments. Body weight of the *Chkb$^{+/+}$* and *Chkb$^{+/-}$* mice showed no difference between groups (Fig. 1A). The *Chkb$^{-/-}$* mice weighed significantly less than their wild type counterparts at all time points. The average body weight of *Chkb$^{-/-}$* mice was 33% to 42% less than that of *Chkb$^{+/+}$* mice at week 6 and week 20, respectively. Dual-Energy X-ray Absorptiometry (DEXA) was utilized to measure bone mineral density and tissue composition (fat mass & lean mass) in 18 weeks old mice (Supplementary Fig. 1A–D). DEXA analysis revealed no significant differences in percent body fat, percent lean mass, bone mineral density and total tissue mass between the wild-type and heterozygous mice. Interestingly, the DEXA analysis showed a 50% decrease in total tissue mass (g) (Supplementary Fig. 1A), 26% decrease in bone mineral density (g/cm$^2$) (Supplementary Fig. 1B), 50% decrease in percent body fat (Figure S1C), and a 20% increase in percent lean mass (Supplementary Fig. 1D) in the knockout mice compared to both the heterozygous and wild-type mice. The findings suggest the Chkb has a role in regulating growth and metabolism.

Forelimb grip strength measurements were performed at three different timepoints and normalized to body weight. The *Chkb$^{-/-}$* mice had significantly lower (less than half) the normalized forelimb strength than wild type mice at all three timepoints (week 6, 12, and 18) (Fig. 1B). Forelimb deformity might have contributed to decreased forelimb grip strength measurements[24]. Another measure of neuromuscular function is the resistance to treadmill running, evaluated as the total distance that each mouse is able to run until exhaustion. The test was performed in all groups at three timepoints (Week 7, 13 and 19). The total distance covered by the wild type mice before exhaustion was similar at all 3 time points (Fig. 1C). There was no significant difference between *Chkb$^{+/+}$* and *Chkb$^{+/-}$* groups, these mice maintained the ability to cover the same total distance before exhaustion (week 7 vs. week 19; non-significant). At week 7, the *Chkb$^{-/-}$* mice showed a basal level of total distance run that was 50% that of the wild type or *Chkb$^{+/-}$* mice. Moreover, the *Chkb$^{-/-}$* mice showed a decline in running performance from week 7 to week 19, with an almost complete inability to run observed by week 19. Gross measurements of neuromuscular strength in whole mice demonstrate that mice heterozygous for

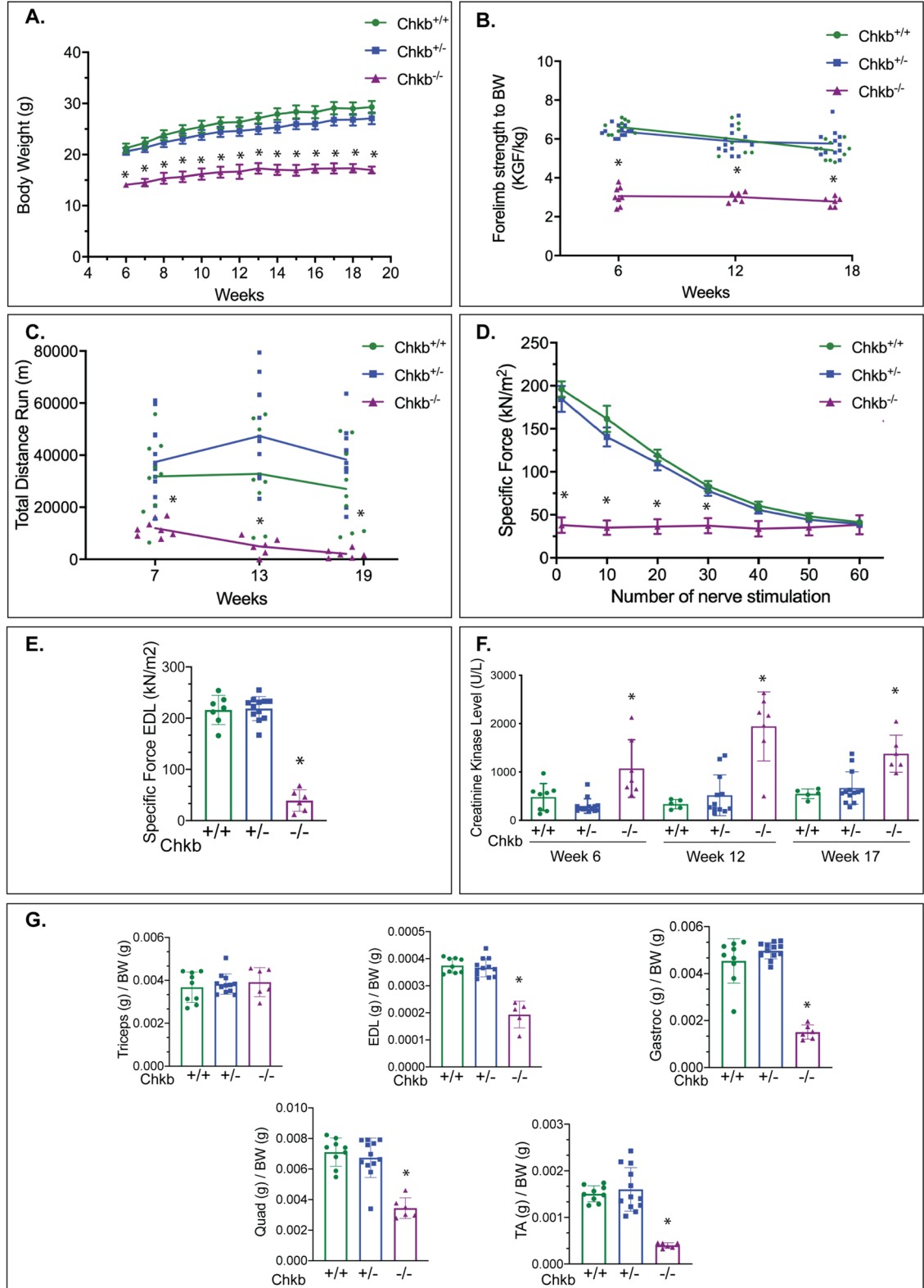

*Chkb* gene display similar phenotypes to wild type mice. Notably, mice lacking both copies of the *Chkb* gene display a significant decrease in overt neuromuscular phenotypes.

To determine if the decreased neuromuscular phenotypes observed in the *Chkb*−/− mice were due to a direct effect on muscle itself, maximal specific force generated by freshly isolated extensor digitorum longus (EDL) muscle from the hindlimb of *Chkb*+/+, *Chkb*+/−, and *Chkb*−/− mice at week 20 was determined. EDL muscle fatigue was measured with 60 isometric contractions for 300 ms each, once every 5 sec, at 250 Hz. There was no significant difference between wild type and heterozygous *Chkb* mice in terms of specific force decrease during fatigue and specific force generation, (Fig. 1D, E). *Chkb*−/− mice displayed a specific EDL force that was 10% that of *Chkb*+/+ or *Chkb*+/− mice. In addition, *Chkb*−/− mice were at maximally fatigued levels on the first stimulation, whereas it took 60 muscle

**Fig. 1 Choline kinase-deficient mice display hallmark muscular dystrophy Phenotypes. A** Body weight was recorded each week at similar times over the entire duration of phenotyping experiment for $Chkb^{+/+}$, $Chkb^{+/-}$, and $Chkb^{-/-}$ mice. **B** Grip strength measurements were performed at 3 different timepoints and normalized to body weight (BW). **C** Total distance run during an exhaustion test for all experimental groups at 3 different timepoints. **D** Loss in muscle force as a result of repeated contractions of EDL muscles by direct stimulation of the nerve for each genotype. **E** Maximal specific force generated by freshly isolated extensor digitorum longus (EDL) muscle for each genotype. **F** Serum creatine kinase (CK) level measurements of 6, 12, and 17-week-old $Chkb^{+/+}$, $Chkb^{+/-}$ and $Chkb^{-/-}$ mice. **G** Muscle weights normalized to body weight of left triceps, EDL, gastrocnemius (Gastroc), quadriceps (Quad), and TA at week 20. All values are expressed as means ± SEM; For **A**, $n = 9$ ($Chkb^{+/+}$), 13 ($Chkb^{+/-}$) and 8 ($Chkb^{-/-}$) mice per group. For **B**, $n = 9$ ($Chkb^{+/+}$), 12 ($Chkb^{+/-}$) and 8 ($Chkb^{-/-}$) mice per group. For **C**, $n = 9$ ($Chkb^{+/+}$), 13 ($Chkb^{+/-}$) and 8 ($Chkb^{-/-}$) mice per group. For **D**, $n = 7$ ($Chkb^{+/+}$), 13 ($Chkb^{+/-}$) and 7 ($Chkb^{-/-}$) mice per group. For **E**, $n = 7$ ($Chkb^{+/+}$), 12 ($Chkb^{+/-}$) and 6 ($Chkb^{-/-}$) mice per group. For **F**, $n = 8$ ($Chkb^{+/+}$), $n = 14$ ($Chkb^{+/-}$) and $n = 7$ ($Chkb^{-/-}$) mice per group. For **G**, $n = 9$ ($Chkb^{+/+}$), $n = 12$ ($Chkb^{+/-}$) and $n = 6$ ($Chkb^{-/-}$) mice per group. For **A**, one-way ANOVA with Tukey's multiple comparison test, $p < 0.0001$ (week 6 to week 19). For **B** one-way ANOVA with Tukey's multiple comparison test, $p < 0.0001$ (week6, week 12, and week 18). For **C**, one-way ANOVA with Tukey's multiple comparison test, $p = 0.0002$ (week6), $p < 0.0001$ (week12 and week 18). For **D** one-way ANOVA with Tukey's multiple comparison test, $p < 0.0001$ (1, 10 and 20 nerve stimulations), $p = 0.0002$ (30 nerve stimulations) and $p = 0.0172$ (40 nerve stimulations). For **E** one-way ANOVA with Tukey's multiple comparison test, $p < 0.0001$. For **F** one-way ANOVA with Tukey's multiple comparison test, $p = 0.00016$ (week6), $p < 0.0001$ (week 12) and $p = 0.0145$ (week17). For **G** one-way ANOVA with Tukey's multiple comparison test, $p = 0.7151$ (for Triceps), $p < 0.0001$ (for EDL, Gastroc, Quad, and TA). *$p < 0.01$ vs. all the other groups. Source data are provided as a Source Data file.

stimulations for $Chkb^{+/+}$ or $Chkb^{+/-}$ to be similarly fatigued. Hindlimb muscle from $Chkb^{-/-}$ mice produce less force, and are much more easily fatigued, than that of wild type or $Chkb$ heterozygous mice.

The level of circulating creatinine kinase (CK), a biomarker of sarcolemmal injury, was determined in $Chkb^{+/+}$, $Chkb^{+/-}$ and $Chkb^{-/-}$ mice longitudinally at three timepoints (week 6, 12, and 17). No significant change in the serum level of CK was observed in $Chkb^{+/-}$ heterozygous mice when compared to the wild type. At all time points, CK activity was 2–3 fold higher in $Chkb^{-/-}$ null mice than that of wild type mice (Fig. 1F). To evaluate the presence of atrophy at week 20 in forelimb and hindlimb of Chkb deficient mice, muscle weights of triceps, EDL, gastrocnemius, quadriceps and TA were measured and normalized to total body weight (Fig. 1G). Consistent with the previously reported rostra-caudal pattern of muscular dystrophy, the normalized forelimb muscle weight (triceps) was similar among groups but there was a significant reduction in the normalized hindlimb muscles weights (EDL, gastrocnemius, quadriceps, and TA) in $Chkb^{-/-}$ mice (Fig. 1F) compared to both $Chkb^{+/+}$ and $Chkb^{+/-}$ mice.

Consistent with earlier reports[5], histological analysis of muscles from 20 weeks old $Chkb^{-/-}$ mice revealed extensive muscle wasting and extreme fatty infiltration in hindlimb; to evaluate the early signs of muscular dystrophy we performed histology on cross sections of hindlimb (gastrocnemius) and forelimb (triceps) muscles from 25-day old $Chkb^{+/+}$, $Chkb^{+/-}$ and $Chkb^{+/-}$ mice (Supplementary Fig. 2A-C). Picro-Sirius Red (PSR) staining showed the presence of fibrotic areas in hindlimb muscles from $Chkb^{-/-}$ mice and to lesser degree in $Chkb^{+/-}$ mice (Supplementary Fig. 2A). Anti-laminin staining of the muscles revealed early signs of muscular dystrophy including a large variation among fiber sizes, rounded shape of atrophic fibers, and increased central nucleation in dystrophic hindlimb muscle from $Chkb^{-/-}$ mice (Supplementary Fig. 2B). H&E staining of the hindlimb muscles showed areas of inflammatory cell infiltration only in hindlimb muscle from $Chkb^{-/-}$ mice. Histological examination of the forelimb skeletal muscle from $Chkb^{-/-}$ mice confirmed that the disease is extremely mild in the forelimbs.

Like humans[8,10], mice with one functional copy of the $Chkb$ gene do not possess any obvious overt muscle dysfunction, whereas mice that are homozygous null for functional copies of the $Chkb$ gene display hallmark muscular dystrophy phenotypes.

**Chka protein expression is inversely correlated with the rostro-caudal gradient of severity in Chkb-mediated muscular dystrophy.** Consistent with the rostral-to-caudal nature of $Chkb$ associated muscular dystrophy, transmission electron micrographs of 115-day old $Chkb^{-/-}$ mice show extensive injury in hindlimb (quadriceps and gastrocnemius) but not the forelimb (triceps) (Fig. 2A, B). $Chkb$ encodes choline kinase b, the first enzymatic step in the synthesis of PC, the most abundant phospholipid in eukaryotic membranes. A second choline kinase, Chka is present in mouse (and human) tissues. We investigated whether the lack of dystrophic phenotypes in $Chkb^{+/-}$ mice, and the rostro-caudal gradient of muscular dystrophy in $Chkb^{-/-}$ muscle, can be explained by compensatory changes in Chkb or Chka protein levels using western blot. In $Chkb^{+/-}$ mice, there was a ~50% decrease in Chkb protein detected in both the forelimb and hindlimb muscles of $Chkb^{+/-}$ mice compared to wild type (Fig. 2C, D). There was no change in Chka protein level in hindlimb muscle of $Chkb^{+/-}$ mice compared to wild type, and a small but statistically insignificant increase in Chka level in forelimb muscle.

In $Chkb^{-/-}$ mouse forelimb or hindlimb muscle, Chkb protein expression was undetectable consistent with the allele not producing Chkb protein. In forelimb muscle from $Chkb^{-/-}$ mice there was a compensatory upregulation of Chka protein expression to almost 3-fold that observed in wild type mice. In contrast, in hindlimb muscle from $Chkb^{-/-}$ mice Chka protein expression was decreased to less than 10% that observed in wild type mice. A compensatory level of Chka protein expression inversely correlates with the rostro-caudal gradient of severity in $Chkb^{-/-}$ associated muscular dystrophy.

Congenital muscular dystrophies associated with $CHKB$ loss-of-function mutations display distinct enlarged (megaconial) mitochondria in the peripheries of muscle fiber and absence of mitochondria in the centers. The distinct characteristic of muscle pathology originally helped the researchers to connect mouse $CHKB$ pathology to human $CHKB$ mutations[8]. However, the dynamic changes in mitochondrial morphology have not been studied before. To understand the temporal development of morphological changes in mitochondria in hindlimb muscle of $Chkb^{-/-}$ mice, we used standard transmission electron microscopy (TEM) stereological methods[25]. The results show that at 12 days of age, the size of mitochondria increased (6.2% ± 0.5 vs 11.4% ± 1.6; $P < 0.01$; wild type vs $Chkb^{-/-}$) while the number of mitochondria (17.3 ± 2.6 vs 16.3 ± 2.1) and cristae density (21.6 ± 2 vs. 23.3 ± 1.9) remained the same. At 60 days of age, the number of mitochondria (18.1 ± 7.6 vs 1.8 ± 0.5; $P < 0.01$) and the cristae density (27.7 ± 1.9 vs. 5.8 ± 1.2; $P < 0.01$) decreased significantly while the size did not change (7.3% ± 0.2 vs 8.2% ± 2.5). At the early stages of Chkb muscular dystrophy, there is an increase in mitochondrial size but not number or morphology. As the disease progresses, the increase in mitochondrial size remains,

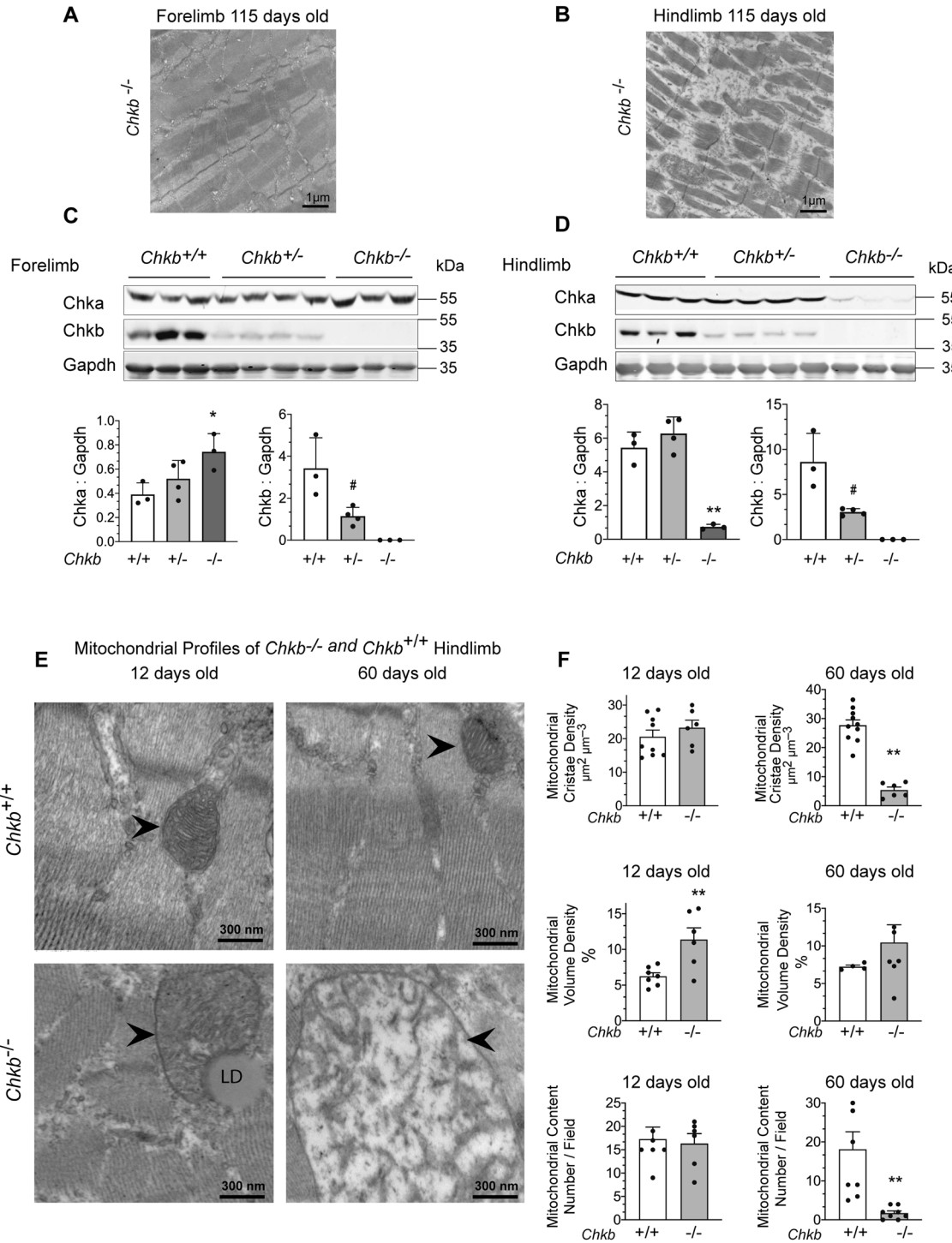

however, the number of mitochondria, and the cristae within mitochondria, decrease (Fig. 2E, F).

**Loss of Chkb activity exerts a major effect on neutral lipid abundance.** PC synthesis is integrated with the synthesis of other major phospholipid classes, as well as AcCa, fatty acids, and the neutral lipids diacylglycerol and triacylglycerol. Lipidomics was used to determine if complete loss of Chkb function, and the associated upregulation of Chka in the forelimb but not hindlimb muscle of $Chkb^{-/-}$ mice, differentially altered lipid metabolism. The levels of the major glycerophospholipids, neutral lipids, and

AcCa in hindlimb and forelimb muscle isolated from 12-day old and 30-day old $Chkb^{+/+}$ and $Chkb^{-/-}$ mice were quantified.

In the forelimb and hindlimb muscle of both 12-day old and 30-day old $Chkb^{-/-}$ mice, the level of PC was the same as wild-type mice (Fig. 3A, B and Supplementary Fig. 3A, B). This is consistent with previous work that determined that there was no difference in PC mass in the $Chkb^{-/-}$ hindlimb, liver, brain, kidney, or heart[5]. Importantly, we determined that in 12-day old $Chkb^{-/-}$ mice the largest change observed was a 15-fold increase in AcCa level (although no change in fatty acyl composition) in hindlimb muscle, and to a lesser degree (~2-fold increase) in forelimb, compared to their wild type littermates.

**Fig. 2 Chka protein expression is inversely correlated with the rostro-caudal gradient of severity in Chkb-mediated muscular dystrophy.** Transmission electron microscopy (TEM) appearance of **A** forelimb (triceps) and **B** hindlimb (quadriceps) of 115-day old $Chkb^{-/-}$ mice showing extensive injury in hindlimb not the forelimb. **A**, **B** are representative of 3 mice per group with similar appearance. Scale bar = 1 μm. Western blot of **C** forelimb (triceps) and **D** hindlimb (quadriceps) samples from three distinct (lanes 1–3) $Chkb^{+/+}$, four distinct (lanes 4–7) $Chkb^{+/-}$ and three distinct (lanes 8–10) $Chkb^{-/-}$ mice probed with anti-Chka, anti-Chkb, and anti-Gapdh antibodies. Bottom: densitometry of the WB data shows the ratio of Chka and Chkb to Gapdh. Chka signal is not significantly different in forelimb and hindlimb samples from $Chkb^{+/-}$ mice compared to the wild type. Chka is upregulated in forelimb muscles and downregulated in hindlimb muscles from $Chkb^{-/-}$ mice. Chkb signal is decreased in hindlimb and forelimb muscle samples of $Chkb^{+/-}$ mice and is absent in muscle samples of $Chkb^{-/-}$ mice. Values in **C** and **D** are means ± SD; For **C** and **D**, $n = 3$ independent $Chkb^{+/+}$, 4 independent $Chkb^{+/-}$ and 3 independent $Chkb^{-/-}$ mice per group. For **C**, left, one-way ANOVA with Tukey's multiple comparison test, $p = 0.04335$. For **C**, right, two-sided student's $t$-test, $p = 0.02842$. For **D**, left, one-way ANOVA with Tukey's multiple comparison test, $p = 0.00011$. For **D**, right, two-sided student's $t$-test, $p = 0.01553$. **E** Transmission electron microscopy (TEM) appearance of the mitochondrial profile of hindlimbs from 12-day old and 60 days old wild-type ($Chkb^{+/+}$) and Chkb-deficient ($Chkb^{-/-}$) mice. For **E**, images are representative of 3 mice per group with similar appearance. Scale bar = 300 nm. **F** At 12 days of age hindlimbs from wild type and $Chkb^{-/-}$ mice had the same number of mitochondria per imaged field however, the volume density of the $Chkb^{-/-}$ mitochondria was increased and the cristae density was preserved. At 115 days of age, $Chkb^{-/-}$ mitochondria were fewer in number, had markedly reduced cristae density, and were much larger in size. The increased size of the mitochondria at this age accounts for the preserved volume density. Values in **F** are means ± SEM. For **F**, top, $n = 9$ (12 days old $Chkb^{+/+}$), $n = 6$ (12 days old $Chkb^{-/-}$), $n = 10$ (60 days old $Chkb^{+/+}$) and $n = 6$ (60 days old $Chkb^{-/-}$) muscle sections from 3 independent mice per genotype for each timepoint. For **F**, middle, $n = 7$ (12 days old $Chkb^{+/+}$), $n = 6$ (12 days old $Chkb^{-/-}$), $n = 5$ (60 days old $Chkb^{+/+}$), $n = 7$ (60 days old $Chkb^{-/-}$) muscle sections from 3 independent mice per genotype for each timepoint. For **F**, bottom, $n = 7$ (12 days old $Chkb^{+/+}$), $n = 6$ (12 days old $Chkb^{-/-}$), $n = 8$ (60 days old $Chkb^{+/+}$) and $n = 8$ (60 days old $Chkb^{-/-}$) muscle sections from 3 independent mice per genotype for each timepoint. Two-sided student's $t$-test. For (**F**, top), $p = 0.39308$, $p = 0.0000$. For **F**, middle, $p = 0.0077$, $p = 0.427594$. For **F**, bottom, $p = 0.78406$, $p = 0.00273$. * and # $p < 0.05$, **$p < 0.01$. Source data are provided as a Source Data file.

The second largest change in 12-day old mice was a 10-fold increase in the level of cardiolipin (CL) in hindlimb muscle that was not present in forelimb muscle of $Chkb^{-/-}$ mice. Phosphatidylethanolamine (PE) and phosphatidylinositol (PI) levels were also slightly increased (~1.5 fold) in both forelimb and hindlimb muscles of 12-day old $Chkb^{-/-}$ mice. The large changes in lipid levels in hindlimb muscle, versus forelimb, of $Chkb^{-/-}$ mice are consistent with the rostral-to-caudal nature of the muscular dystrophy observed in these mice.

Considering the progressive nature of the disease, we tracked the changes in the lipid profile in the hindlimb of 30-day old $Chkb^{-/-}$ mice, when muscle injury is more pronounced. In sharp contrast to 12-day old mice, AcCa and CL levels were no longer increased and were at the same level as wild type mice. Instead, there was a 12-fold increase in the neutral storage lipid TG and a 3-fold increase in its precursor DG in the hindlimb samples of $Chkb^{-/-}$ mice (Fig. 3C, D). PE and PS levels were 2-3-fold higher in the hindlimb samples from 30-day old $Chkb^{-/-}$ mice compared to wild type littermates. Specific to affected muscle in $Chkb^{-/-}$ mice, there is an early temporal shift resulting in a 12 to 15-fold increase in CL and AcCa that becomes resolved as the disease progresses with the lipid profile transitioning to a 12-fold increase in TG.

As AcCa levels are many times higher than wild type mice in the early stage of $Chkb^{-/-}$ muscular dystrophy, this implies that the affected muscles are unable to use fatty acids for energy production by mitochondrial β-oxidation. As $Chkb^{-/-}$ muscular dystrophy progresses, the affected muscles appear to adapt to this inability to consume fatty acids by transitioning toward energy storage indicated by the large increase in TG.

To examine early ultrastructural pathological changes, and to further explore the the nature of the accumulation of TG in affected muscle as $Chkb^{-/-}$ muscular dystrophy progresses, we performed transmission electron microscopy (TEM) on hindlimb muscles from 12- and 115-day old mice. A closer examination of hindlimb muscle TEM images from 12-day old mice revealed early signs of disrupted sarcomeres, as well as a small increase in the abundance of cytoplasmic lipid droplets, consistent with the small (2-fold) and statistically insignificant increase in TG in hindlimb muscle we observed using lipidomics. Interestingly, these lipid droplets were located mainly adjacent to enlarged mitochondria (Fig. 3E). Detailed quantification of

randomly imaged lipid droplets in hindlimb muscle from 12-day old $Chkb^{-/-}$ mice revealed that 81% were associated with mitochondria (Fig. 3E and Supplementary Fig. 3D). In 115-day old $Chkb^{-/-}$ mice, cytoplasmic lipid droplets increased substantially in size (Fig. 3E).

We also evaluated TG accumulation in muscle using confocal microscopy by staining hindlimb muscle sections of 30-day old $Chkb^{-/-}$ mice with BODIPY 493/503 (Fig. 3F). Concanavalin A dye conjugate (CF™ 633) and Dapi were used to stain membrane (red) and nucleus (blue) respectively. Consistent with our TEM and lipidomics results, BODIPY-stained lipid droplets were noticeably more frequent and larger in $Chkb^{-/-}$ hindlimb muscles compared to the wild type littermates. The same pattern of lipid droplet staining was observed using Nile red staining (Supplementary Fig. 3C). Consistent with the increase in TG level, there was a 3.5 and 3.3 fold increase in gene expression of the TG synthesis enzymes DGAT1 and DGAT2 in $Chkb^{-/-}$ hindlimb muscle compared to the wild type (Supplementary Fig. 3E) consistent with an increase in TG synthesis resulting in the increase in lipid droplets observed.

**Changes in expression of peroxisome proliferator-activated receptors (Ppars) and target genes reinforce the observed changes in lipid levels in $Chkb^{-/-}$ hindlimb muscle.** Our lipidomic data imply a temporal shift from the use of fatty acids for energy to the storage of lipids in affected muscle tissues of $Chkb^{-/-}$ mice. To further investigate this metabolic shift, we examined the expression of Ppars in affected muscles of 30 day old $Chkb^{+/+}$, $Chkb^{+/-}$ and $Chkb^{-/-}$ mice. Ppars are master regulators of lipid metabolism[26,27]. The endogenous ligands for Ppars are fatty acids and their derivatives. There are three Ppar members, each encoded by distinct genes, designated Ppara, Pparb/d and Pparg. Ppara and Pparb/d primarily regulate the expression of genes required for fatty acid oxidation, with Pparb/d also regulating genes required for mitochondria biogenesis. Pparg is primarily expressed in adipose tissue and regulates insulin sensitivity and glucose metabolism[26]. Using reverse transcription (RT) qPCR, we determined that the expression of Ppara and Pparb/d were 4-fold and 6-fold lower, while Pparg was 2-fold higher, in the hindlimb muscle of 30-day old mice $Chkb^{-/-}$ mice compared to wild type (Fig. 4A). Consistent with RT qPCR results, assessment of Ppar protein levels by western

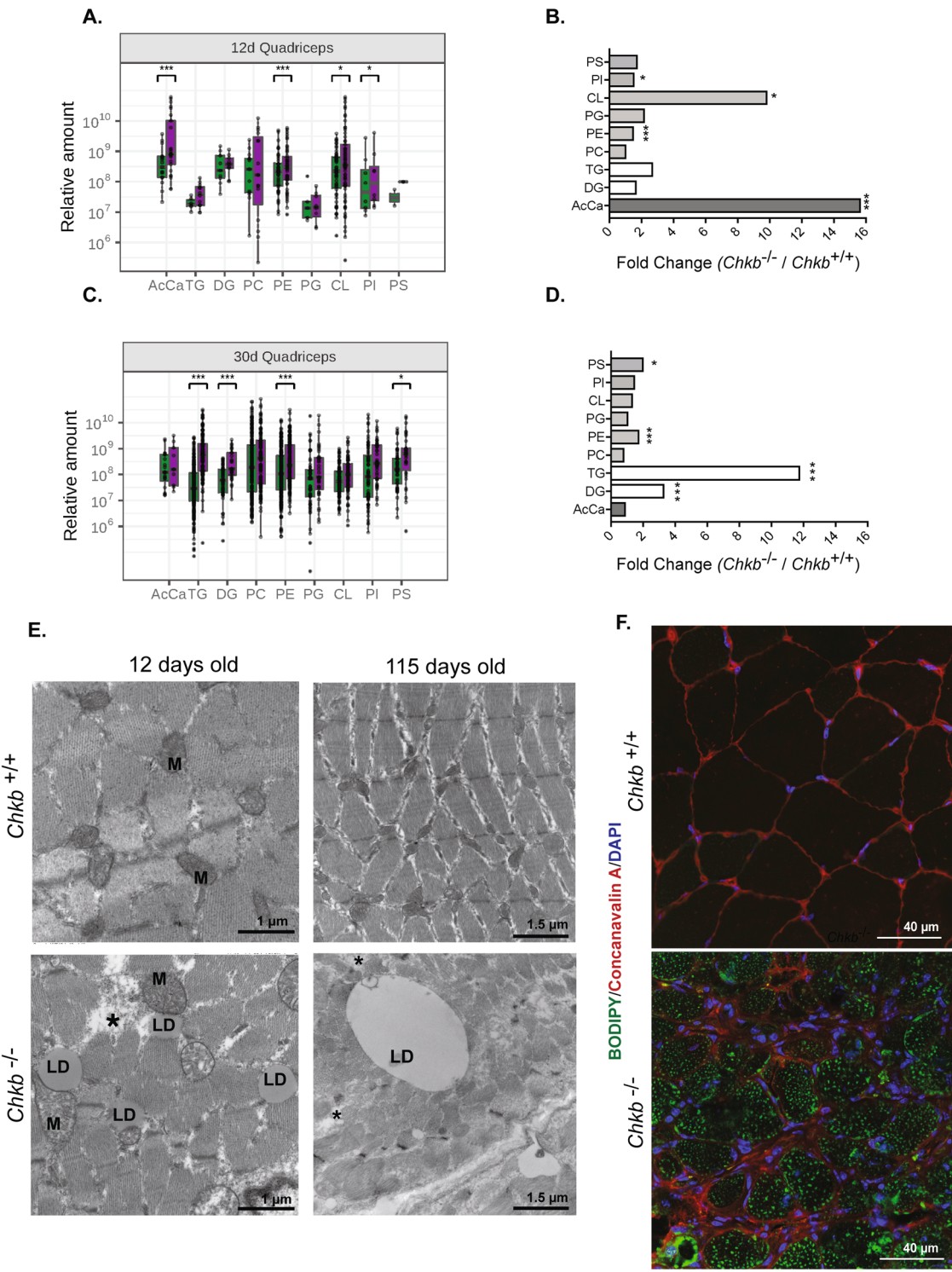

blot show decreased Ppara and Pparb/d protein expression, and an increase in Pparg protein expression, in Chkb deficient hindlimb muscle compared to wild type (Fig. 4B). Ppar protein levels did not change in Chkb deficient forelimb muscle compared to $Chkb^{+/+}$ (Supplementary Fig. 4A-B) indicating that the Ppar changes are isolated to affected muscle.

To further evaluate and validate how the Ppar pathway contributes to the lipid metabolic changes observed in $Chkb^{-/-}$ mice, we utilized a microarray of 82 Ppar regulated genes, along with 4 housekeeping genes, to assess transcriptional changes in the hindlimb muscle of 30-day old mice. As expected, there was

no change in the expression of the 4 housekeeping genes. For Ppar receptors to bind to Ppar response elements in gene promoters, Ppars form obligate heterodimers with Retinoid X receptors (Rxr). There are three members of the Rxr family, Rxra, Rxrb, and Rxrg, and their expression was reduced 8-, 5-, and 16-fold in hindlimb muscle of $Chkb^{-/-}$ mice compared to wild type (Fig. 4C, D). Ppar and Rxr heterodimers are bound to DNA with coactivator molecules[28] and the expression of each co-activator was also decreased from 2.8- to 14.4-fold compared to wild type (Fig. 4C, D). The several fold decrease in expression of the Ppars, as well as their obligate co-receptors, aligns well with the observed

**Fig. 3 Loss of Chkb activity exerts a major effect on neutral lipid abundance.** Comparison of expression levels of major lipids between the $Chkb^{+/+}$ and $Chkb^{-/-}$ mice. The analysis was performed on **A, B**, 12-day old hindlimb (quadriceps), and **C, D**, 30 days old hindlimb (quadriceps) samples. Each point is an individual lipid species (specific fatty acid composition). **B, D** Summary of fold change and statistical tests performed on major glycerophospholipids. **A, C** Boxplots bounds (hinges) correspond to the 25th and 75th percentiles of the data with the boxplot center corresponding to the median value. The upper/lower whiskers extend from the hinges to the largest/smallest value no further than 1.5 times the interquartile range away from the hinges. For **A–D**, $n = 3$ independent $Chkb^{+/+}$ and 3 independent $Chkb^{-/-}$ mice. Pairwise Wilcoxon signed rank test with Bonferroni correction was used to determine the significance of a median pair-wise fold-increase in lipid amounts at an overall significance level of 5%. All statistical tests were two-sided. As the Bonferroni correction is fairly conservative, significant differences are reported at both pre-correction (*) and post-correction (***) significance levels. AcCa, acylcarnitine; TG, triacylglycerol; DG, diacylglycerol; PC, phosphatidylcholine; PE, phosphatidylethanolamine; PG, phosphatidylglycerol; PI, phosphatidylinositol; PS, phosphatidylserine. For (**A, B**), $p = 0.0002$ (AcCa), $p = 0.0.2500$ (DG), $p = 0.0625$ (TG), $p = 0.1748$ (PC), $p = 0.0000$ (PE), $p = 0.1250$ (PG), $p = 0.0070$ (CL), $p = 0.0078$ (PI), $p = 1$ (PS). For (**C, D**), $p = 1$ (AcCa), $p = 0.0001$ (DG), $p = 0.0000$ (TG), $p = 0.5059$ (PC), $p = 0.0000$ (PE), $p = 0.6720$ (PG), $p = 0.2162$ (CL), $p = 0.1495$ (PI), $p = 0.0179$ (PS). **E** Transmission electron microscopy (TEM) appearance of the hindlimb muscle samples (quadriceps) of $Chkb^{+/+}$ and $Chkb^{-/-}$ mice at 12 days and 115 days of age. Images are representative of 3 mice per group with similar appearance. LD = Lipid droplets. M = Mitochondria. * = Disrupted sarcomeres. Scale bar = 1.5 µm. **F** Quadriceps muscle sections of 30 days old $Chkb^{+/+}$ and $Chkb^{-/-}$ mice were fixed and stained with BODIPY-493/503 to visualize LDs (Green). Concanavalin A dye conjugate (CF™ 633) and DAPI were used to stain membrane (Red) and nucleus (Blue) respectively. Images are representative of 3 mice per group with similar results. Scale bar = 40 µm. Source data are provided as a Source Data file.

changes in lipid profiles we observed in affected muscle of $Chkb^{-/-}$ mice that predict a decreased capacity to import and use fatty acids by mitochondria for β-oxidation.

Among Ppar associated genes, the expression of 44 genes was decreased statistically significantly ($P < 0.05$) by at least 2-fold in $Chkb^{-/-}$ mice, while 8 genes were upregulated at least 2-fold (Fig. 4C–F, Supplementary Fig. 5, Supplementary Table 1 and Supplementary Table 2). Carnitine palmitoyltransferase 1b (Cpt1b), the major muscle isoform of Cpt, is involved in the carnitine shuttle as it catalyzes the conversion of cytoplasmic long-chain fatty acyl-CoA and carnitine into AcCa that are translocated across the inner mitochondrial membrane for subsequent mitochondrial fatty acid β-oxidation. The expression of Cpt1b was decreased 7.9-fold in affected muscle of $Chkb^{-/-}$ mice. In addition, the expression of enzymes required for mitochondrial fatty acid β-oxidation were also decreased by several folds in the affected muscle of $Chkb^{-/-}$ mice including several fatty acyl CoA synthases/ligases, fatty acid binding proteins, and fatty acid β-oxidation enzymes.

Ppara and Pparb/d are the major transcriptional reporters that regulate expression of fatty acid metabolizing genes. The many-fold decrease in the expression of these Ppars that was specific to affected muscle, along with their coreceptors and downstream target genes corroborate the lipidomics data that suggest that the major change in lipid metabolism in Chkb mediated muscular dystrophy is an inability to metabolize fatty acids via mitochondrial β-oxidation resulting in shunting of excess fatty acid into TG rich lipid droplets.

**Chkb deficiency results in decreased fatty acid oxidation and increased lipid droplet accumulation in differentiated myocytes in culture.** To address if the observed increase in TG in $Chkb^{-/-}$ mice was due to muscle-specific events or was due to larger physiological changes that then impact muscle physiology, we assessed fatty acid utilization and TG level in primary cultured muscle cells after myoblast differentiation.

We first determined if Chkb deficiency alters differentiation in primary myoblasts. Primary muscle cell cultures were examined for their transition from a single cell proliferative condition to differentiated multinucleated myotubes. During the process of differentiation, mononuclear myoblasts fuse to form myocytes (myotubes), which are large multinucleated cells. We isolated skeletal myoblasts from $Chkb^{+/+}$ and $Chkb^{-/-}$ mice and induced differentiation by switching to low growth factor serum. Representative light micrographs of cultures of dissociated myogenic cells from the skeletal muscle of $Chkb^{+/+}$ and $Chkb^{-/-}$

mice at 0, 4 and 8 days after switching to differentiation media show a similar degree of myotube formation (Fig. 5A). Chkb deficiency resulted in a compensatory upregulation of Chka gene expression as well as a significant increase in markers of myocyte injury, namely Icam1 and Tgfb1[29] (Fig. 5C). We calculated the fusion index, which is nuclei distribution, to determine the extent of myotube differentiation, by immunofluorescence staining. There was no difference between the $Chkb^{+/+}$ and $Chkb^{-/-}$ cells in terms of the percentage of nuclei within the myotubes, the average number of nuclei in each myotube, or the distribution of nuclei in myotubes (Fig. 5D). Loss of Chkb function does not appear to affect gross myoblast differentiation.

To test whether Chkb deficiency and reduced expression of Ppars and target genes in skeletal muscle affected the fatty acid oxidation capacity of these cells, we measured the oxygen consumption rates by primary $Chkb^{+/+}$ and $Chkb^{-/-}$ myocytes at day 4 and 8 of differentiation using a Seahorse XF24 extracellular flux analyzer. The same number of primary myoblasts were seeded into different wells of the same Seahorse plate and oxygen consumption was determined using glucose/glutamine/pyruvate as energy source, or fatty acid as energy source, at days 4 and 8 of differentiation. At day 4 of differentiation maximal respiration driven by glucose/glutamine/pyruvate was significantly increased in $Chkb^{-/-}$ myocytes compared to the wild type (Fig. 5E). Contrary to this, there was a slight but significant decrease in maximal respiration driven by long-chain fatty acid as fuel (Fig. 5F). At day 8 of differentiation there was a larger reduction in maximal respiration driven by fatty acids in $Chkb^{-/-}$ myocytes (Fig. 5H) while the maximal respiration driven by other fuels (glucose/glutamine/pyruvate) was similar to wild type (Fig. 5G). $Chkb^{-/-}$ myocytes demonstrate progressively declining mitochondrial fatty acid oxidation capacity.

To assess whether Chkb deficiency and decreased fatty acid utilization modulates TG storage in isolated myotubes, we stained differentiated myotubes with BODIPY 493/503 to visualize neutral lipid droplets. Lipid droplets were noticeably more abundant and larger in Chkb deficient myotubes compared to wild type (Fig. 5I). Quantification of the corrected total cell fluorescence intensity in $Chkb^{-/-}$ myotubes confirmed a 2-fold increase in lipid droplet formation (Fig. 5J). The increase in TG level in differentiated muscle cells isolated from $Chkb^{-/-}$ mice is in line with the increased TG and lipid droplet levels observed in isolated hindlimb muscle from older $Chkb^{-/-}$ mice and implies that the increase in TG in hindlimb muscle due to the loss of Chkb function is a direct effect on lipid metabolism within the muscle cells themselves.

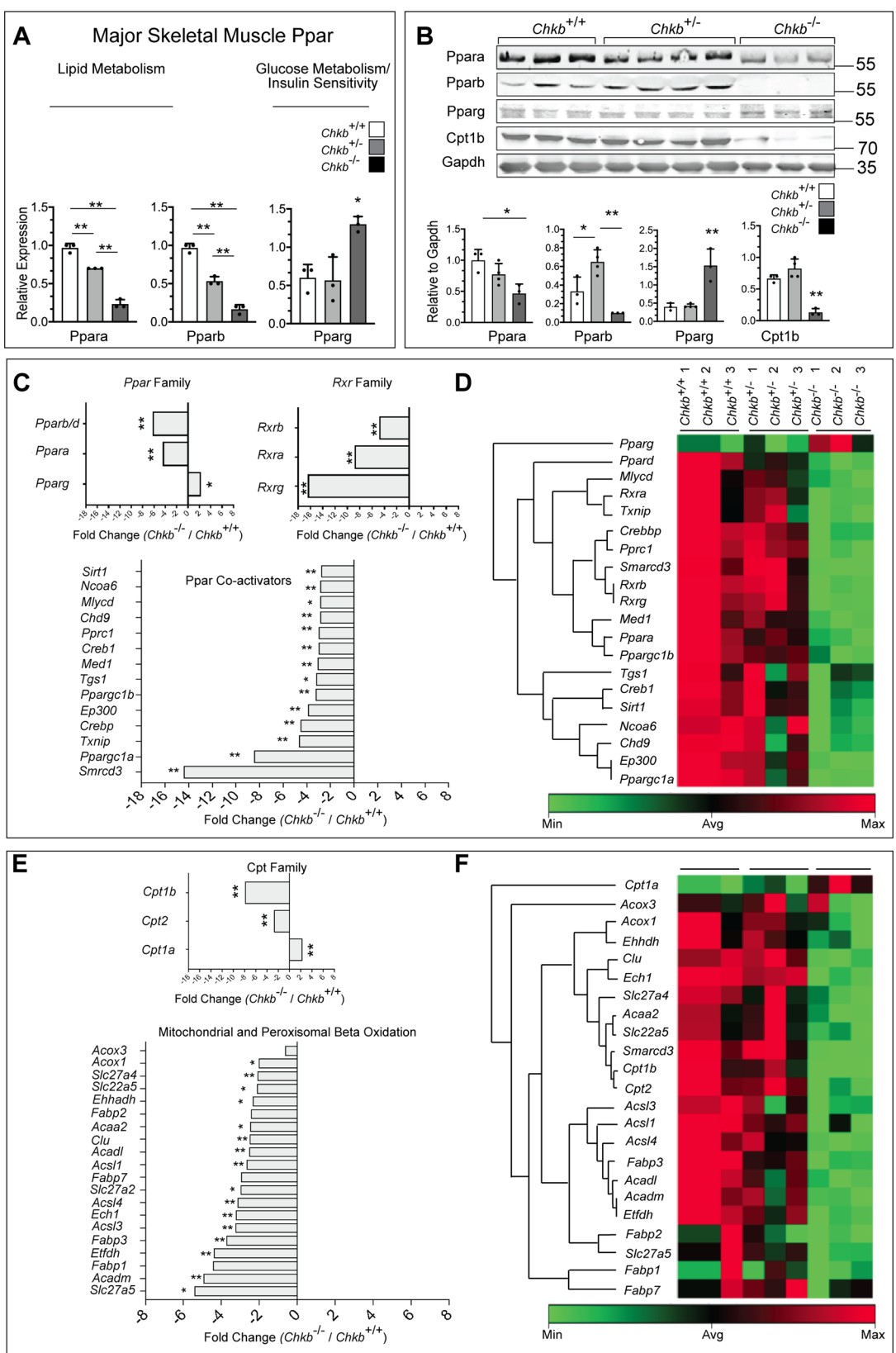

**Ppar activation rescues defective fatty acid utilization and lipid droplet accumulation in differentiated *Chkb*⁻ᐟ⁻ myocytes in culture**. To determine whether the decreased fatty acid utilization and increased lipid droplet accumulation in *Chkb*⁻ᐟ⁻ myocytes is mediated by a Ppar signaling pathway(s) we treated *Chkb*⁻ᐟ⁻ myocytes with the Ppara agonists ciprofibrate and fenofibrate, the pan Ppar agonist bezafibrate, and the Pparb/d specific agonist GW501516 and assessed the capacity of the

**Fig. 4 Chkb regulates the gene expression of the members of the Ppar family as well as Ppar target genes. A** Relative gene expression of the *Ppar* family members. **B** Western blot of hindlimb (quadriceps) samples from three distinct (lanes 1–3) $Chkb^{+/+}$, four distinct (lanes 4–7) $Chkb^{+/-}$ and three distinct (lanes 8–10) $Chkb^{-/-}$ mice probed with anti-Para, anti- Pparb, anti-Pparg, anti-Cpt1b, and anti-Gapdh antibodies. Bottom: densitometry of the western blot data shows the ratio of Ppara, Pparb, Pparg, and Cpt1b to Gapdh. Values are means ± SD. For **A**, $n = 3$ ($Chkb^{+/+}$), $n = 3$ ($Chkb^{+/-}$) and $n = 3$ ($Chkb^{-/-}$) mice per group. For **B**, $n = 3$ ($Chkb^{+/+}$), $n = 4$ ($Chkb^{+/-}$) and $n = 3$ ($Chkb^{-/-}$) mice per group. For **A** one-way ANOVA with Tukey's multiple comparison test, $p < 0.0001$ (Ppara), $p < 0.0001$ (Pparb), $p = 0.0087$ (Pparg). *$p < 0.05$, **$p < 0.01$. For (**B**) one-way ANOVA with Tukey's multiple comparison test, $p = 0.0166$ (Ppara), $p = 0.0014$ (Pparb), $p = 0.0010$ (Pparg), $p = 0.00018$ (Cpt1b). *$P < 0.05$, **$P < 0.01$. **C** Fold-Change (2^ (- Delta Delta CT)) is the normalized gene expression (2^(- Delta CT)) in the Chkb deficient hindlimb sample divided the normalized gene expression (2^ (- Delta CT)) in the control sample. Fold-change values greater than one indicates a positive- or an up-regulation. Fold-change values less than one indicate a negative or down-regulation, and the fold-regulation is the negative inverse of the fold-change. The *p* values are calculated based on a two-sided student's *t*-test of the replicate 2^ (- Delta CT) values for each gene in the $Chkb^{+/+}$ group and $Chkb^{-/-}$ groups. *$p < 0.05$, **$p < 0.01$. $n = 3$ independent samples per group. **D** The clustergram of the *Ppar* family, *Rxr* family and *Ppar* coactivators across three genotypes. **E** Fold-Change (2^ (- Delta Delta CT)) is the normalized gene expression (2^(- Delta CT)) in the Chkb deficient hindlimb sample divided the normalized gene expression (2^ (- Delta CT)) in the control sample. Fold-change values greater than one indicates a positive- or an up-regulation. Fold-change values less than one indicate a negative or down-regulation, and the fold-regulation is the negative inverse of the fold-change. The p values are calculated based on two-sided student's *t*-test of the replicate 2^ (- Delta CT) values for each gene in the $Chkb^{+/+}$ group and $Chkb^{-/-}$ groups. *$p < 0.05$, **$p < 0.01$. $n = 3$ independent samples per group. **F** The clustergram of the *Ppar* family, *Rxr* family and *Ppar* coactivators across three genotypes. Average arithmetic means of the expression of 4 housekeeping genes (*Actb*, *B2m*, *Gusb* and *Hsp90ab1*) were used to normalize the expression of all the studied genes. Source data are provided as a Source Data file.

myocytes to oxidize fatty acid. Quantification of maximal respiration showed that all the Ppar agonists enhanced maximal respiration of $Chkb^{-/-}$ myocytes (Fig. 6A, C, D, F), with the specific Pparb/d agonist (GW501516) able to increase basal respiration in $Chkb^{-/-}$ myocytes to the same level as $Chkb^{+/+}$ myocytes (Fig. 6D, E).

Interestingly, previous work had observed that fibrates could increase *Chka* expression in neuronal cells[30]. We tested the effect of Ppar activation, plus/minus choline supplementation, on *Chka* expression in $Chkb^{-/-}$ myocytes. We found that choline, along with ciprofibrate, GW501516, or bezafibrate treatment resulted in a 2-to-6-fold increase in *Chka* gene expression (Fig. 6G–I). We sought to determine if this increase in Chka expression by Ppar agonists affected the conversion of choline to phosphocholine, the substrate, and product of choline kinases[31–33], and if the addition of choline further enhanced this conversion. Using targeted metabolomic profiling we determined that Chkb deficiency results in the expected accumulation of choline and decrease in phosphocholine in differentiated $Chkb^{-/-}$ myocytes (Fig. 6J, K); and are consistent with previous work that monitored [³H] choline incorporation into Kennedy pathway metabolites in hindlimb muscle of $Chkb^{-/-}$ mice[7]. Ppar activation by bezafibrate normalized the choline and phosphocholine levels in $Chkb^{-/-}$ myocytes. Choline supplementation along with bezafibrate resulted in an even larger increase in phosphocholine level indicating a much higher flux through the choline kinase step (Fig. 6K). Indeed, the level of choline also increased in the presence of both bezafibrate and choline (Fig. 6J) indicating that flux through the Kennedy pathway had saturated its rate-determining step. Ppar agonists increase expression of the *Chka* gene that normalized its substrate and product levels in differentiated $Chkb^{-/-}$ myocytes consistent with the restoration of PC synthesis via the Kennedy pathway.

The ability of Ppar agonists (plus/minus choline) was further assessed with respect to their capacity to resolve of the observed increase in AcCa level in hindlimb of $Chkb^{-/-}$ mice. Consistent with our observation of increased AcCa levels in hindlimb muscles from 12-day old $Chkb^{-/-}$ mice, AcCa levels were 5.4-fold higher in differentiated $Chkb^{-/-}$ myocytes in culture compared to $Chkb^{+/+}$ myocytes (Fig. 6L). Ppar activation, plus/minus choline supplementation, reduced AcCa levels in differentiated $Chkb^{-/-}$ myocytes to near those seen in wild type cells. This observation aligns with our result that Ppar agonist treatment increases the capacity of $Chkb^{-/-}$ myocytes to increase maximal mitochondrial respiration.

The other major change observed in lipid level in affected muscle of $Chkb^{-/-}$ mice was the increase in lipid droplets, TG, and to a lesser extent DG. To determine if Ppar agonists also prevented lipid droplet formation, four days after differentiation $Chkb^{-/-}$ myocytes were treated with bezafibrate or GW501516 for 48 h and on day 6 of differentiation the medium was supplemented with Oleate-BSA (400 μM) overnight. The cells were then labeled with mitotracker® Red CMXRos and stained with BODIPY 493/503 to visualize LDs. LDs were noticeably more abundant and larger in $Chkb^{-/-}$ myocytes compared to the wild type cells. Ppar activation by bezafibrate or GW501516 significantly decreased lipid droplets in Chkb deficient myotubes to a level comparable to wild type (Fig. 7A). Furthermore, analysis by lipidomics determined that TG and DG levels were 1.93 and 1.22-fold higher in differentiated $Chkb^{-/-}$ myocytes compared to wild type and that both TG and DG levels were normalized in $Chkb^{-/-}$ myocytes treated with bezafibrate plus/minus choline supplementation compared to wild type (Fig. 7B). Finally, we sought to determine if Ppar agonists plus/minus choline reduced injury to $Chkb^{-/-}$ myocytes. Choline along with ciprofibrate, bezafibrate, or GW501516 substantively reduced injury marker levels in $Chkb^{-/-}$ myocytes (Fig. 7C).

## Discussion

This study highlights that (1) a change in PC level is not the major metabolic driver behind this inherited muscular dystrophy despite the fact that the genetic defect lies within the sole metabolic pathway for the synthesis of PC in muscle cells, (2) a surprising mechanistic model for the disease in which there are two temporal stages beginning with an inability to use fatty acids for mitochondrial β-oxidation and a compensatory shunting of fatty acids for storge as TG rich lipid droplets, (3) deficient expression of Ppara, and Pparb/d, along with their target genes is the main mechanism for the inability of fatty acids to be used for mitochondrial β-oxidation and their shunting into lipid droplets, and (4) based on the above mechanistic insights we show that that Ppar activation can prevent $Chkb^{-/-}$ myocyte injury as Ppar activators reverse the above metabolic defects while simultaneously increasing the expression of *Chka* (a paralog of *Chkb*), pointing to the use of Ppar activators along with choline (the metabolic precursor for PC synthesis) as a potential therapy for this muscular dystrophy (Fig. 8).

Importantly, we report that instead of a change in PC level, there is an unexpected 12- to 15-fold increase in the levels of the mitochondrial specific lipids CL and AcCa at an early

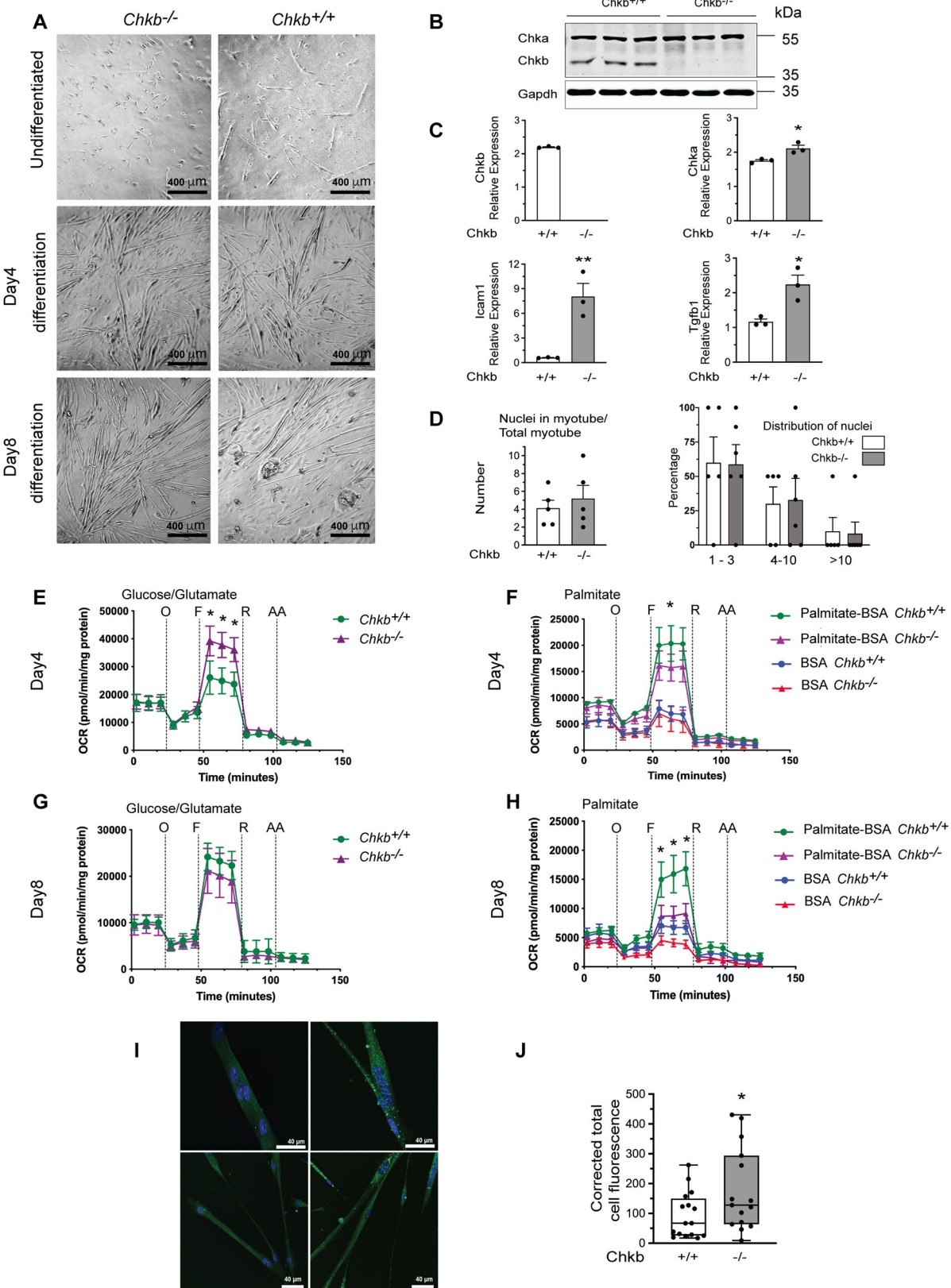

stage of *Chkb* mediated muscular dystrophy in affected muscle of *Chkb*$^{-/-}$ mice. As the disease progresses, AcCa and CL levels return to wild type, and a 12-fold increase in the TG/lipid droplets occurs. The 15-fold increase in AcCa level in affected muscle implied that there was an inability use fatty acids for subsequent fatty acid β-oxidation and this was confirmed by our direct analysis of mitochondrial respiration in *Chkb*$^{-/-}$ myocytes where reduced fatty acid oxidation capacity progressively decreased over time. As *Chkb*$^{-/-}$ mice aged, AcCa level in hindlimb muscle returned to wild type and by 30 days a dramatic 12-fold increase in TG level was observed. The increase in TG level is consistent with impaired use of fatty acids by mitochondrial β-oxidation

**Fig. 5 Chkb deficiency results in decreased fatty acid usage and increased lipid droplet accumulation in differentiated myocytes in culture.**
**A** Representative image of isolated skeletal myoblasts from $Chkb^{+/+}$ and $Chkb^{-/-}$ mice, cultured on Matrigel® coated culture flasks. At day 0, when the cells reached 80% confluency, the medium was replaced by differentiation medium and maintained in differentiation media for up to 8 days. **B** Western blot of differentiated $Chkb^{+/+}$ and $Chkb^{-/-}$ myocytes probed with anti-Chka, anti-Chkb, and anti-Gapdh antibodies. For **A** and **B**, Each experiment was repeated independently 3 times with similar results. **C** RT-qPCR analysis of gene expression in isolated myocytes from $Chkb^{+/+}$ and $Chkb^{-/-}$ mice at day 5 of differentiation. Values are means ± SEM; $n = 3$ independent experiments. Two-sided student's $t$-test, $p = 0.0200$ (Chka), $p = 0.0095$ (Icam1), $p = 0.01944$ (Tgfb1). *$p < 0.05$, **$p < 0.01$. **D** Formaldehyde fixed and immunostained myotubes were categorized into three groups (1 to 3 nuclei, 4 to 10 nuclei, and >10 nuclei per myotube). The number of multinuclear myotubes in the two groups and the distribution of nuclei were calculated to compare differentiation in primary $Chkb^{+/+}$ and $Chkb^{-/-}$ myocytes. For **D** values are means ± SEM. $n = 5$ random images from 3 independent experiments for each group. Two-sided student's t-test did not result in $p$ values less than 0.05. Representative traces of oxygen consumption rates (OCRs) of primary $Chkb^{+/+}$ and $Chkb^{-/-}$ myocytes at day 4 (**E**, **F**) and day 8 (**G**, **H**) of differentiation driven with glucose/glutamine/pyruvate or palmitate as stated. Bovine serum albumin (BSA) alone was used as control for palmitate-BSA complex driven OCRs. Oligomycin(O), FCCP(F), rotenone (R) and antimycin A (AA) were sequentially injected to assess mitochondrial respiratory states. Data are mean ± SD. For **E**, $n = 5$ ($Chkb^{+/+}$) and $n = 5$ ($Chkb^{-/-}$) wells per group. For (**F**), $n = 5$ (BSA $Chkb^{+/+}$), $n = 5$ (BSA $Chkb^{-/-}$), $n = 5$ (Palmitate-BSA $Chkb^{+/+}$), $n = 5$ (Palmitate-BSA $Chkb^{-/-}$). For **G**, $n = 10$ ($Chkb^{+/+}$) and $n = 8$ ($Chkb^{-/-}$) wells per group. For (**H**), $n = 4$ (BSA $Chkb^{+/+}$), $n = 4$ (BSA $Chkb^{-/-}$), $n = 4$ (Palmitate-BSA $Chkb^{+/+}$), $n = 4$ (Palmitate-BSA $Chkb^{-/-}$). For **E–H** Two-sided student's $t$-test. For **E**, during maximal respiration, $p = 0.0062$, $p = 0.0021$, $p = 0.00187$. For **F**, during maximal respiration, $p = 0.0857$, $p = 0.0421$, $p = 0.0526$. For **G**, during maximal respiration, $p = 0.1242$, $p = 0.1139$, $p = 0.0713$. For **H**, during maximal respiration, $p = 0.01231$, $p = 0.0072$, $p = 0.0038$. *$p < 0.05$. **I** Isolated primary myocytes from $Chkb^{+/+}$ and $Chkb^{-/-}$ mice were fixed 5 days after differentiation and stained with BODIPY-493/503 to visualize LDs (green). DAPI was used to stain nucleus (blue). **J** The corrected total cell fluorescence intensity of lipid droplets was significantly enhanced in $Chkb^{-/-}$ myotubes. Box plots in **J** show median, quartiles (boxes), and range (whiskers). For **J**, $n = 16$ ($Chkb^{+/+}$) and $n = 15$ ($Chkb^{-/-}$) myotubes. For **J** Two-sided student's $t$-test. For **J**, $p = 0.0463$. *$p < 0.05$. **I** and **J** are representative of 3 independent experiments with similar results. Source data are provided as a Source Data file.

resulting in a shunting of fatty acids from energy source to energy storage[34]. A recent study demonstrated that lipid droplets adjacent to mitochondria have reduced fatty acid oxidation capacity and instead promote lipid droplet/TG synthesis[35], interestingly we observed that ~80% of the lipid droplets from $Chkb^{-/-}$ hindlimb muscles were closely associated with mitochondria. Indeed, as we determined that there was impaired mitochondrial fatty acid oxidation and increased TG/lipid droplets in $Chkb^{-/-}$ hindlimb muscle, our data adds a disease context to the determination that peri-mitochondrial lipid droplet accumulation is due to shunting of fatty acids away from energy generating β-oxidation and toward lipid storage.

The unexpected transitions in defective fatty acid metabolism prompted us to examine if Ppars, master regulators of lipid metabolism that are regulated by fatty acids, were affected and we determined that there was a substantial decrease expression of *Ppara* and *Pparb/d*, as well as their target genes including those required for AcCa uptake into mitochondria and their subsequent β-oxidation. Based on this observation, went on to show that Ppar agonists reversed the lipid metabolic defects and resulted in an ability to use of fatty acids for mitochondrial β-oxidation and their administration prevented the accumulation of lipid droplets. The same Ppar agonists also increased *Chka* expression in $Chkb^{-/-}$ affected muscle which would enable PC synthesis to now occur in these cells. This prompted us to test if the addition of the Kennedy pathway precursor choline, along with the Ppar agonists, could substantively prevent injury to $Chkb^{-/-}$ affected muscle and this was indeed the case. Our study has determined how lipid metabolism is specifically altered in affect $Chkb^{-/-}$ affected muscle, how to reverse these metabolic alterations, and determined a potential treatment for *CHKB* mediated muscular dystrophy.

A perplexing aspect of *Chkb* mediated muscular dystrophy is the fact that PC level does not change despite the fact that the sole genetic defect is inactivation of a gene that encodes the first step of the PC biosynthetic pathway, and what causes the rostral-to-caudal gradient of the disease. We propose that the rostral-to-caudal gradient of the disease is due to compensatory effects modulated by a second choline kinase isoform, *Chka*. In hindlimb muscle of $Chkb^{-/-}$ mice we observed a marked reduction in Chka protein level, while conversely in the forelimb of $Chkb^{-/-}$

mice there was a compensatory upregulation of Chka. Consistent with this explanation is our finding that enhancing Chka expression via the addition of Ppar protected against *Chkb*-mediated muscular cell injury. This is also consistent with previous work that demonstrated that the viral delivery of the *Chka* gene improved the dystrophic phenotypes of $Chkb^{-/-}$ mice with comparable potency to rescue by delivery of $Chkb$[36]. One caveat to the above is the fact that PC level was not different between the forelimb and hindlimb muscle of $Chkb^{-/-}$ mice, nor was it different between wild type and $Chkb^{-/-}$ mice at any stage of the disease. This suggests that PC supply must be able to be replenished at a step downstream of choline kinase. We propose that PC level does not change as PC can be replenished via exogenous PC supply. PC is imported into cells from serum via low-density lipoproteins (LDL), and enhanced expression of scavenger receptor-B1 (SR-B1) and low-density lipoprotein receptor (LDLR) was previously observed in muscle of $Chkb^{-/-}$ mice, both of which would be expected to enhance the uptake of plasma PC[7]. We propose that the expected decrease in the level of PC in hindlimb muscle of $Chkb^{-/-}$ mice, due to inactivation of the *Ckkb* gene and downregulation of *Chka* gene expression, is not observed as this can be compensated for by increased PC uptake from serum. These predictions are consistent with the changes in muscle function along the rostral-to-caudal gradient in $Chkb^{-/-}$ mice but require further research to confirm.

The increase in the mitochondrial specific phospholipid CL in the early stage of *Chkb* mediated muscular dystrophy is also quite telling. Our TEM of mitochondria in affected muscle during the early stage of disease revealed a similar number of mitochondria with intact cristae in compared to wild type, however, there was a substantive increase in large mitochondria in affected muscle of $Chkb^{-/-}$ mice. We propose that the large increase in CL in affected muscle of $Chkb^{-/-}$ mice in the early stage of the disease is mainly driven by the increase in mitochondrial size. As the mice aged the level of CL decreased and had returned to that of wild type by 30 days. At 30 days, mitochondrial size was still increased, however, the number of mitochondria, as well as their cristate (where the bulk of CL resides) were substantively decreased, providing a reasonable explanation for CL mass being reduced to wild type level as the $Chkb^{-/-}$ mice aged. This is consistent with observations of mitochondria in $Chkb^{-/-}$ mice that had with

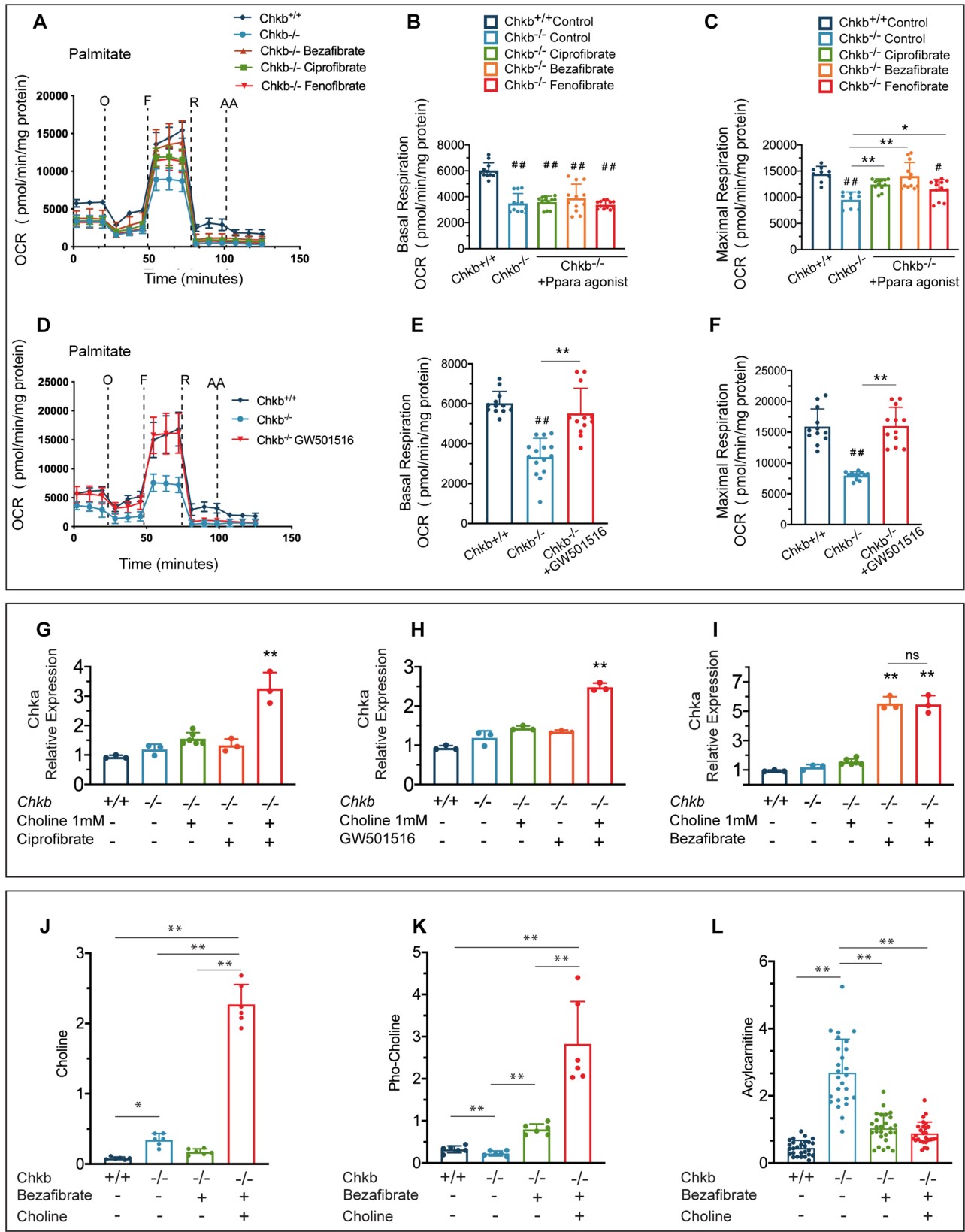

advanced disease[6,37], where similar changes in mitochondrial morphological features and numbers were observed.

## Methods

**Mouse strains**. All animal procedures were approved by the Dalhousie University's Committee on Laboratory Animals in accordance with guidelines of the

Canadian Council on Animal Care Guide to the Care and Use of Experimental Animals (CCAC, Ottawa, ON, Canada: vol. 1, 2nd ed., 1993; vol. 2, 1984). Mice were housed in ventilated microisolator cages. Corn cob bedding was used with Enviro-dri as enrichment. A dawning-dusking light cycle (13-hour light/11-hour dark cycle) was used. Temperature set point was 22 degrees C (+/− 2 degrees) and humidity set point was 50% (+/−10 %). Chkb mutant mice in C57BL/6J background were a kind gift of Professor Gregory A. Cox and were originally generated

**Fig. 6 Ppar activation rescues defective fatty acid utilization and normalizes the choline and phosphocholine levels in differentiated $Chkb^{-/-}$ myocytes in culture. A** Representative kinetic graph of the fatty acid oxidation of primary $Chkb^{-/-}$ myocytes at day 7 of differentiation. The cells were treated with or without ciprofibrate (50 µM), bezafibrate (500 µM) or fenofibrate (25 µM) in the medium 72 h prior to measurement. Values are means ± SEM; $n = 4$ for each group. Each experiment was repeated independently 3 times with similar results. Quantification of basal respiration and maximal respiration which quantifies maximal electron transport activity induced by the chemical uncoupler FCCP. Values are means ± SEM; For **B**, $n = 12$ ($Chkb^{+/+}$), $n = 12$ ($Chkb^{-/-}$), $n = 12$ ($Chkb^{-/-}$ Bezafibrate), $n = 12$ ($Chkb^{-/-}$ Fenofibrate), $n = 12$ ($Chkb^{-/-}$ Cipofibrate) wells per group. One-way ANOVA with Tukey's multiple comparison test, $p < 0.0001$. $\#\#p < 0.01$ vs $Chkb^{+/+}$ group. For **C** $n = 9$ ($Chkb^{+/+}$), $n = 9$ ($Chkb^{-/-}$), $n = 12$ ($Chkb^{-/-}$ Bezafibrate), $n = 12$ ($Chkb^{-/-}$ Fenofibrate) and $n = 12$ ($Chkb^{-/-}$ Cipofibrate). One-way ANOVA with Tukey's multiple comparison test, $p < 0.0001$. $*p < 0.05$, $**p < 0.01$, $\#p < 0.05$ vs $Chkb^{+/+}$ group, $\#\#p < 0.01$ vs $Chkb^{+/+}$ group. **D** Representative kinetic graph of the fatty acid oxidation of primary $Chkb^{-/-}$ myocytes at day 7 of differentiation. The cells were treated with or without specific pparb/d agonist GW501516 (2.5 µM) in the medium 72 h prior to measurement. Values are means ± SEM; For **D**, $n = 4$ ($Chkb^{+/+}$), $n = 5$ ($Chkb^{-/-}$) and $n = 4$ ($Chkb^{-/-}$ GW501516) wells per group. Each experiment was repeated independently 3 times with similar results. **E**, **F** Quantification of basal respiration and maximal respiration which quantifies maximal electron transport activity induced by the chemical uncoupler FCCP. Values are means ± SEM; For **E**, $n = 12$ ($Chkb^{+/+}$), $n = 15$ ($Chkb^{-/-}$), $n = 12$ ($Chkb^{-/-}$ GW501516). One-way ANOVA with Tukey's multiple comparison test, $p < 0.0001$. $**p < 0.01$, $\#\#p < 0.05$ vs $Chkb^{+/+}$ group. RT-qPCR analysis of $Chka$ gene expression in differentiated $Chkb^{+/+}$, $Chkb^{-/-}$ and $Chkb^{-/-}$ myocytes treated with or without Ciprofibrate 50 (µM) (**G**), GW501516 (2.5 µM) (**H**) or bezafibrate (500 µM) (**I**) in the medium for 48 h on day 4 of differentiation with or without choline (1 mM) supplementation. For **G–I**, $n = 3$ independent samples per group. One-way ANOVA with Tukey's multiple comparison test, $p < 0.0001$. $**P < 0.01$. **J–L** Targeted metabolomic profiling of $Chkb^{+/+}$, $Chkb^{-/-}$ and $Chkb^{-/-}$ myocytes treated with bezafibrate (500 µM) for 48 h on day 4 of differentiation with or without choline (1 mM) supplementation. Ppar activation increases phosphocholine (p-Choline) level (**K**) and normalizes acylcarnitine (AcCa) level (**L**) in differentiated $Chkb^{-/-}$ myocytes. Values are means ± SEM. For **J** and **K**, $n = 6$ samples for each group. One-way ANOVA with Tukey's multiple comparison test. $p < 0.0001$. Each experiment was repeated independently 3 times with similar results. For **L**, $n = 15$ AcCa species from 3 independent samples for each group. One-way ANOVA with Tukey's multiple comparison test. $P < 0.0001$. Each experiment was repeated independently 3 times with similar results. $*p < 0.05$ and $**p < 0.01$. Source data are provided as a Source Data file.

at the Jackson Laboratory (Bar Harbor, Maine, USA)[5]. Male $Chkb^{+/-}$ mice on the C57BL/6J background were crossed with female $Chkb^{+/-}$ on the same background to generate $Chkb^{+/+}$, $Chkb^{-/-}$ and $Chkb^{+/-}$ littermates. The mutation identified in $Chkb^{-/-}$ mice is a 1.6 kb genomic deletion between exon 3 and intron 9 that results in expression of a truncated mRNA and the absence of Chkb protein expression[5].

**Mouse genotyping.** The mutation identified in $Chkb^{-/-}$ mice is a 1.6 kb genomic deletion between exon 3 and intron 9[5]. AccuStart™ II Mouse Genotyping Kit (Beverly, MA, USA) was used to extract DNA from ear punches and to perform PCR analysis. A single genotyping program was used to amplify both the wild type $Chkb$ allele between exons 5 and 9 and the truncated $Chkb$ allele between exons 2 and 10. The primers used for genotyping were purchased from Integrated DNA Technologies (Coralville, IA, USA). The primer sequences to genotype wild type are Forward Primer: 5′-GTG GGT GGC ACT GGC ATT TAT −3′; Reverse Primer: 5′-GTT TCT TCT GTT CCT CTT CGG AGA-3' (amplicon size 753 bp).

The primer sequences to genotype the mutants are: Forward Primer: 5′-TAC CCA CGT ACC TCT GGC TTT T −3′ Reverse Primer: 5′-GCT TTC CTG GAG GAC GTG AC 3′(amplicon size 486 bp). For each mouse, one PCR reactions was performed using both the primer sets. If two bands were observed, the mouse was characterized as a heterozygous.

**In vivo grip strength and fatigability measurements.** Forelimb grip strength was measured using a grip strength meter (Columbus Instruments, Columbus, OH, USA) at 3 time points (6, 12, 18 weeks old) as previously described[38]. All mice were acclimated for a period of five consecutive days before testing. For each time point, Force measurements were collected in the morning hours over a 5-day period, with maximum values for each day over this period averaged to obtain absolute GSM values (Kgf) or normalized to BW (recorded on the first day of testing) for normalized GSM values (Kgf/kg). For the treadmill exhaustion assay, mice were subjected to an enforced running paradigm that tests the resistance level of fatigue in mice. The exhaustion test was performed at 3 time points (7, 13, 19 weeks old) in each group. Groups of mice were made to run on a horizontal treadmill for 5 min at 5 m/min, followed by an increase in the speed of 1 m/min each minute. The total distance run by each mouse until exhaustion was measured. Exhaustion was defined as the inability of the mouse to continue running on the treadmill for 30 s, despite repeated gentle stimulation.

**Ex vivo force measurement.** At the end of the in vivo phase (Week 19), mice were deeply anesthetized with ketamine and xylazine (80 and 10 mg/kg). The extensor digitorum longus (EDL) muscle of the right hindlimb was removed for comparison of Ex vivo force contractions between groups as previously described[39,40]. Briefly, the EDL muscle was securely tied with braided surgical silk at both tendon insertions to the lever arm of a servomotor/force transducer (model 305B) (Aurora Scientific, Aurora, Ontario, Canada) and the proximal tendon was fixed to a stationary post in a bath containing buffered Ringer solution (composition in mM: 137 NaCl, 24 NaHCO₃, 11 glucose, 5 KCl, 2 CaCl₂, 1 MgSO₄, 1 NaH₂PO₄ and 0.025 turbocurarine chloride) maintained at 25 °C and

bubbled with 95% $O_2$−5% $CO_2$ to stabilize pH at 7.4. At optimal muscle length, the maximal force developed was measured during trains of stimulation (300 milliseconds, ms) with increasing frequencies up to 250 Hz or until the highest plateau was achieved. The force generated to obtain the highest plateau was used to determine specific force (maximal force normalized to cross-sectional area of the muscle). Finally, the muscle was subjected to a fatigue protocol consisting of 60 isometric contractions for 300 ms each, once every 5 s. The frequency at which the EDL muscles were stimulated is 250 Hz. The force was recorded every 10th contraction during the repetitive contractions and again at 5 and 10 min afterward to measure recovery.

**Creatine kinase (CK) serum levels.** CK was determined from serum taken from blood samples withdrawn by cheek bleed at 3 time points (5, 10, and 15 weeks old). Blood was centrifuged for 3000 g for 10 min at 4 °C to obtain the serum. CK determination was performed by standard spectrophotometric analysis, using a CK diagnostic kit (Cat. no. C7522-450, PONITSCIENTIFIC, Canton, MI, USA) according to the manufacturer instructions.

**Total RNA isolation, cDNA generation, and quantitative real-time RT qPCR.** Isolated tissue samples were incubated overnight in pre-chilled RNAlater® (Cat. no. R0901, Sigma-Aldrich, Ontario, Canada) at 4 °C. Tissues were then homogenized in TRIzol reagent (Cat. no. 15596026, Invitrogen, MA, USA) and total RNA was isolated according to the manufacturer's protocol. Nine hundred nanograms of total RNA was reverse transcribed using High-Capacity cDNA Reverse Transcription Kit® (Cat. no. 4368814, Applied Biosystems, MA, USA). Quantitative real-time RT-PCR assays were performed on the Bio-Rad CFX96 Touch Real-Time PCR Detection (Bio-Rad®, California, USA) System using TaqMan™ Fast Advanced Master Mix (Cat. no. 4444557) and TaqMan™Gene Expression Assays (Cat. no. 4331182, ThermoFisher Scientific) for Chka (RRID: Mm00442759_m1), Chkb Cpt1b (Exon boundary7-8)(RRID: Mm01308102_g1), Cnsk2a2 (RRID: Mm01243455_m1), Cpt1b (RRID: Mm00487191_g1), Gapdh (RRID: Mm99999915_g1), Icam1 (RRID: Mm00516023_m1), Ppara (RRID: Mm00440939_m1), Ppard (RRID: Mm00440940_m1), Pparg (RRID: Mm00440940_m1) and Tgfb1 (RRID: Mm011778820_m1). Reactions were run in triplicate. For qPCR analysis, Light Cycler® 96 Instrument Software, Version 1.1.1 was used.

**Microarray analysis of Ppar targets.** Mature RNA was isolated using Qiagen RNeasy Plus Mini Kit (Cat. no 74134) according to the manufacturer's instructions. RNA quality was determined using a spectrophotometer and was reverse transcribed using a cDNA conversion kit. The cDNA was used on the real-time RT2 Profiler PCR Array (QIAGEN, Cat. no. PAMM-149Z) in combination with RT2 SYBR® Green qPCR Mastermix (Cat. no. 330529). CT values were exported to an Excel file to create a table of CT values. This table was then uploaded on to the data analysis web portal at http://www.qiagen.com/geneglobe. Samples were assigned to controls and test groups. CT values were normalized based on a/an Manual Selection of reference genes. The data analysis web portal calculates fold change/regulation using delta delta CT method, in which delta CT is calculated between

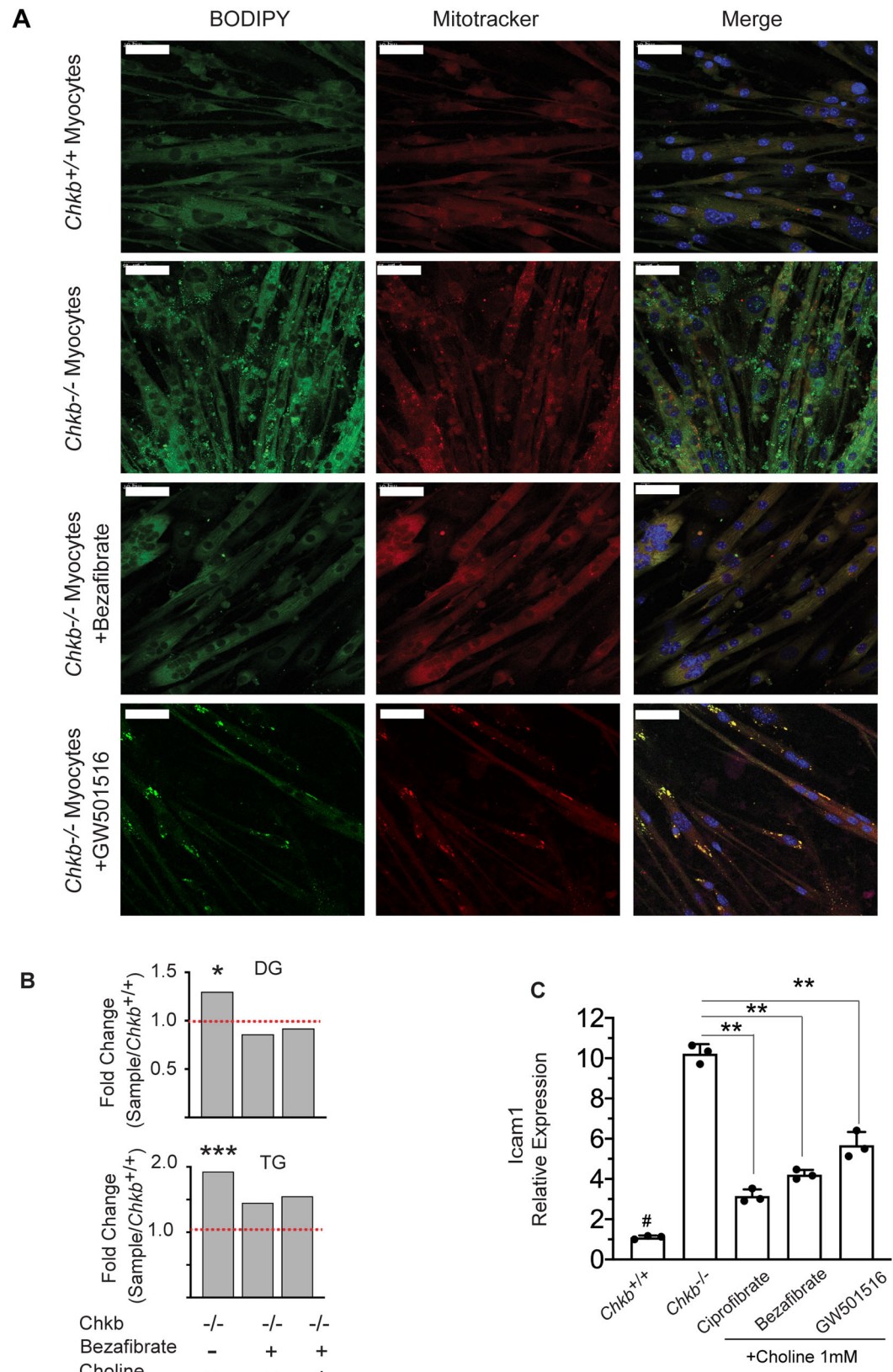

gene of interest (GOI) and an average of reference genes (HKG), followed by delta-delta CT calculations (delta CT (Test Group)-delta CT (Control Group)). Fold Change is then calculated using 2^ (-delta delta CT) formula.

**Lipid extraction**. We performed lipid extractions using the modified Bligh and Dyer extraction for LC-MS analysis of lipids protocol[41]. All reagents were of LC-MS grade. Briefly, the muscle tissue (~10 mg) was homogenized with a steel bead in 1 ml of cold 0.1 N HCl: methanol (1:1, v/v) using a TissueLyser II instrument (Qiagen) set at 30 strokes/s for 2–4 min. Based on protein quantification results, all samples were adjusted to the final concentration of 700 µg/ml and spiked with 10 µl of internal standard (Avanti Polar Lipids Inc; Catalog Number-330707). 500 µl of chloroform was added to each sample, vortexed for 30 min, and centrifuged to separate phases (5 min at 6000 rpm). The bottom organic phase was transferred

**Fig. 7 Ppar activation normalizes lipid droplet accumulation and decrease injury in differentiated Chkb$^{-/-}$ myocytes in culture. A** To study the effect of Ppar agonist treatments on exogenous fatty acid utilization and storage, 4 days after differentiation, myocytes were treated with or without bezafibrate (500 μM) or GW501516 (2.5 μM) in the medium for 48 h. On day 6 of differentiation the medium with or without drugs was supplemented with Oleate-BSA (400 μM) overnight. On day 7 of differentiation cells were labeled with mitotracker® Red CMXRos (50 nM) for 30 min, washed with PBS, fixed and stained with BODIPY-493/503 to visualize LDs (Green). DAPI was used to stain nucleus (Blue). The images are representative of 3 independent experiments with similar results. Scale bar = 50 μm. **B** Summary of fold change and statistical tests performed on DG and TG. n = 3 independent samples per group. Pairwise Wilcoxon signed rank test with Bonferroni correction was used to determine the significance of a median pair-wise fold-increase in lipid amounts at an overall significance level of 5%. All statistical tests were two-sided. As the Bonferroni correction is fairly conservative, significant differences are reported at both pre-correction (*) and post-correction (***) significance levels. TG, triacylglycerol; DG, diacylglycerol. $p = 0.0069$ (DG) and $p = 0.0007$(TG). **C** RT-qPCR analysis of Icam1 gene expression in differentiated Chkb$^{++}$ myocyte and Chkb$^{-/-}$ myocytes treated with ciprofibrate (50 μM), bezafibrate (500 μM) or GW501516 (2.5 μM) in the presence of 1 mM choline for 48 h on day 4 of differentiation. Values are means ± SD; n = 3 independent experiments. One-way ANOVA with Tukey's multiple comparison test; $p < 0.0001$. #$p < 0.01$ (Chkb$^{++}$) vs all the other groups. **$p < 0.01$. Source data are provided as a Source Data file.

into a new Eppendorf and dried under a nitrogen stream. Samples were stored at −80 °C until ready for analysis.

**UHPLC method for lipid analysis.** The Accucore C30 column (250 × 2.1 mm I.D., particle size: 2.8 μm) was obtained from ThermoFisher Scientific (ON, Canada). The mobile phase system consisted of solvent A (acetonitrile: H2O 60:40 v/v) and solvent B (isopropanol: acetonitrile: water 90:10:1 v/v) both containing 10 mM ammonium formate and 0.1% formic acid. C30-RPLC separation was carried out at 30 °C (column oven temperature) with a flow rate of 0.2 mL/min, and 10 μL of the lipid extraction suspended in the mobile phase solvents mixtures (A: B, 70:30%) was injected onto the column. The following system gradient was used for separating the lipid classes and molecular species: 30% solvent B for 3 min; then solvent B increased to 50% over 6 min, then to 70% B in 6 min, then kept at 99% B for 20 min, and finally the column was re-equilibrated to starting conditions (30% solvent A) for 5 min prior to each new injection. For standardization, we used the SPLASH® Lipidomix® Mass Spec Standard as which includes deuterium labeled forms of all the major lipid classes.

**High resolution tandem mass spectrometry and lipidomics.** Lipid analyses were carried out using a Q-Exactive Orbitrap mass spectrometer controlled by X-Calibur software 4.0 (ThermoScientific, MO, USA) with an acquisition HPLC system. The following parameters were used for the Q-Exactive mass spectrometer - sheath gas: 40, auxiliary gas: 5, ion spray voltage: 3.5 kV, capillary temperature: 250 °C; mass range: 200–2000 m/z; full scan mode at a resolution of 70,000 m/z; top-1 m/z and collision energy of 35 (arbitrary unit); isolation window: 1 m/z; automatic gain control target: 1e5. The instrument was externally calibrated to 1 ppm using ESI negative and positive calibration solutions (ThermoScientific, MO, USA). Tune parameters were optimized using a mixture of lipid standards (Avanti Polar Lipids, Alabama, USA) in both negative and positive ion mode Thermo Scientific™ LipidSearch™ software version 4.2 was used for lipid identification and quantitation. First, the individual data files were searched for product ion MS/MS spectra of lipid precursor ions. MS/MS fragment ions were predicted for all precursor adduct ions measured within ±5 ppm. The product ions that matched the predicted fragment ions within a ±5 ppm mass tolerance was used to calculate a match-score, and those candidates providing the highest quality match were determined. Next, the search results from the individual positive or negative ion files from each sample group were aligned within a retention time window (±0.2 min) and the data were merged for each annotated lipid.

**Data cleanup and statistical analysis of lipids.** Thermo Scientific™ LipidSearch™ software version 4.2 was used for lipid identification and quantification. An in-house script written in R (version 4.0.2) was used for data QC analysis, normalization, and plotting. The data was filtered to exclude any peak concentration estimates with a signal to noise ratio (SNR parameter) of less than 2.0 or a peak quality score (PQ parameter) of less than 0.8. If this exclusion resulted in the removal of two observation within a biological triplicate, the remaining observation was also excluded. The individual concentrations were then gathered together by lipid identity (summing together the concentration of multiple mass spectrometry adducts where these adducts originated from the same molecular source and averaging together biological replicates) and grouped within the broader categories of AcCa, TG, DG, PC, PE, PG, CL, PI, PS. The result was nine groups containing multiple lipid concentrations corresponding to specific lipid identities, which were then compared between wild type and KO samples using a (paired, non-parametric) Wilcoxon signed-rank test at an overall significance level of 5% (using the Bonferroni correction to account for the large number of tests performed). As the Bonferroni correction is fairly conservative, significant differences are reported at both pre-correction (*) and post-correction (***) significance levels.

**Nile red 550 / 640 nm, BODIPY 493/503 nm, and nuclei staining of muscle tissue.** Quadriceps and gastrocnemius muscles were embedded in Optimal Cutting

Temperature™ (Sakura Finetek, Torrence, CA), and were frozen in cooled iso-pentane in liquid nitrogen and stored at −80 °C. Frozen sections (7 μm thick) were thaw-mounted on SuperFrost Microscope slides (Microm International, Kalamazoo, MI) and air-dried. Tissue sections were then fixed in 4% (w/v) paraformaldehyde for 15 min and incubated with Concanavalin A CF® Dye Conjugates CF™633 (50–200 μg/mL) for 20 min followed by incubation with either Nile red solution in PBS (0.5 μg/mL) or BODIPY 493/503 for 15 min. The sections were then washed for 5 times with PBS, each time for 15 min, and mounted using ProLong™ Gold Antifade Mountant with DAPI (Thermo Scientific™, Cat. no. P36931) and cured overnight in the dark. Slides were observed under a confocal microscope (Leica TCS SP8 with LIGHTNING) using excitation wavelength 633 for Concanavalin A, 550 nm for Nile red, 448 nm for BODIPY, and 405 for DAPI.

**Primary myoblast isolation, culture and differentiation.** We followed a protocol outlined in Shahini et al[42]. for isolation of myoblast by enabling the outgrowth of these cells from muscle tissue fragments of Chkb$^{+/+}$ and Chkb$^{-/-}$ mice. Briefly, the mice were euthanized via CO2, were sprayed with 70% ethanol, and transferred to a sterile hood. The forelimb and hindlimb muscles were removed, finely minced into small pieces, and transferred to a 50 ml conical tube. 1 ml enzymatic solution of PBS containing collagenase type II (500 U/mL), collagenase D (1.5 U/mL), dispase II (2.5 U/mL), and CaCl2 (2.5 mM) was added to the tube. The muscle mixture was placed in a water bath at 37 °C for 60 min with agitation every 5 min. The suspension was centrifuged for 10 min at 300 g. Following centrifugation, the supernatant was removed and discarded, and the pellet was resuspended in proliferation medium. Proliferation medium composed of high glucose Dulbecco's Modified Eagle Medium (DMEM, Gibco, Grand Island, NY), 20% fetal bovine serum (FBS, Atlanta Biologicals, Flowery Branch, GA), 10% horse serum (HS, Gibco), 0.5% chicken embryo extract (CEE, Accurate Chemical, and Scientific, Westbury, NY), 2.5 ng/mL bFGF (ORF Genetics, Iceland), 10 μg/mL gentamicin (Gibco), and 1% Antibiotic-Antimitotic (AA, Gibco), and 2.5 μg/mL plasmocin prophylactic (Invitrogen, San Diego, CA). The re-suspended pellet containing small pieces of muscle tissue was plated on matrigel-coated flasks at 10-20% surface coverage and incubated at 37 °C and 5% CO2 to allow attachment of the tissues to the surface and subsequent outgrowth and migration of myoblasts. The myogenic cell population was further purified with one round of pre-plating on collagen-coated dishes to isolate fibroblasts from myoblasts. To induce differentiation into multinucleated myotubes, the cells were seeded at 10000 cells/cm$^2$ on plastic coverslip chambers coated with Matrigel, and the medium was replaced by differentiation medium containing DMEM with high glucose and 5% HS.

**BODIPY 493/503 and nuclei staining of primary myocytes.** Isolated skeletal myoblasts were cultured on Matrigel® coated Glass chamber slides (Thermo Scientific™, Cat. no. 154534) and differentiated into myocyte. 3 days after differentiation, the cells were washed two times with PBS and fixed in 4% (w/v) paraformaldehyde for 15 min. The cells were washed with PBS for 10 min and incubated with BODIPY solution in PBS for 15 min, at room temperature on a shaker. The cells were then washed for 3 times with PBS, each time for 15 min, and mounted using ProLong™ Gold Antifade Mountant with DAPI (Thermo Scientific™, Cat. no. P36931) and cured overnight in the dark. Slides were observed under a confocal microscope (Leica TCS SP8 with LIGHTNING) using excitation wavelength 448 nm for BODIPY and 405 for DAPI. Images were converted to 8-bit and the total corrected cellular fluorescence for the green channel was measured in random 100 cells per group using FIJI (NIH; ImageJ 2.0.0-rc-69/1.52n) software. The total corrected cellular fluorescence (TCCF) = integrated density−(area of selected cell × mean fluorescence of background readings), was calculated and compared between groups[43].

To study the effect of Ppar agonist treatments on exogenous fatty acid utilization and storage, 4 days after differentiation, myocytes were treated with or without ciprofibrate (50 μM), bezafibrate (500 μM), and fenofibrate (25 μM) in the medium for 48 h. On day 6 of differentiation the medium with or without drugs

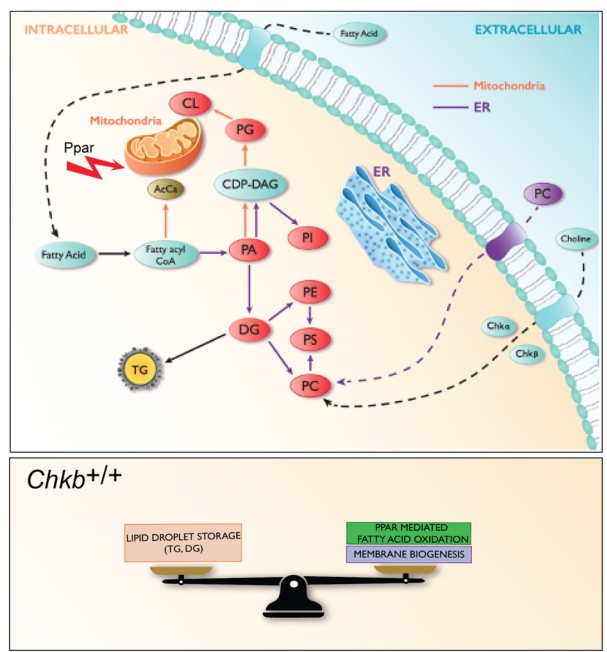

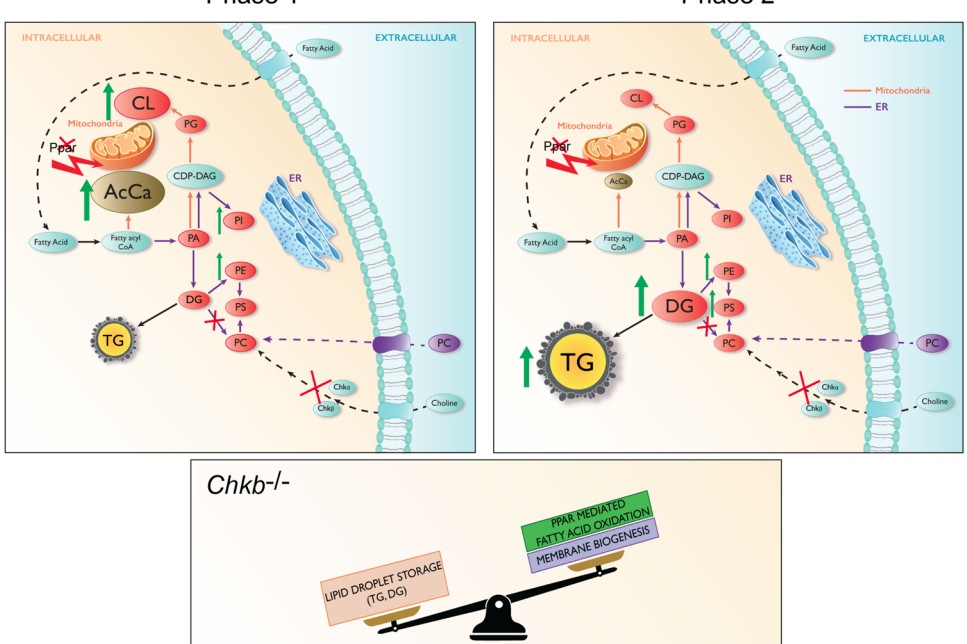

**Fig. 8 Defective fatty acid utilization and lipid metabolism in Chkb⁻/⁻ hindlimb muscle.** In muscles from *Chkb⁺/⁺* mice there is a balance between storage of fatty acid as triacylglycerol (TG) and usage of fatty acids either as an energy source by mitochondria β-oxidation or membrane phospholipid synthesis. In hindlimb muscles from *Chkb⁻/⁻* mice, an inability to consume DG for PC synthesis results in an imbalance between storage and usage of fatty acids. Although the cells are able to increase PC uptake from plasma to compensate for defective PC synthesis, this genetic defect in PC synthesis drives large fluctuations in lipid metabolism. At an early stage of *Chkb* mediated muscular dystrophy (Phase 1), there is a 12- to 15-fold increase in the levels of the mitochondrial specific lipids CL and AcCa; the large increase in CL reflects the increase in mitochondrial size at this stage of the disease. The increase in AcCa level is likely due to the inability to consume DG for PC synthesis, resulting in an accumulation of its precursor-fatty acid which the cell attempts to consume via mitochondria β-oxidation, however there is a concomitant decrease in Ppar mediated expression of genes required for fatty acid conversion to AcCa for its import into mitochondria and consumption by β-oxidation. As the disease progresses (Phase 2), CL level returns to wild type (probably as a result of damaged mitochondrial inner membrane), and a 12-fold increase in the storage lipid TG occurs due to an inability consume AcCa and the shunting of fatty acids into storage lipid droplets.

was supplemented with Oleate-BSA (400 µM) overnight. On day 7 of differentiation cells were labeled with mitotracker® Red CMXRos (50 nM) for 30 min, washed with PBS, fixed, and stained with BODIPY 493/503 as described earlier.

**Seahorse analysis of mitochondrial function**. Oxygen consumption rate (OCR) was measured as previously described[44] using a Seahorse XF24 extracellular flux analyzer (Seahorse Biosciences, North Billeric, MA, USA). Agilent Seahorse Wave Desktop software version 2.6 was used to transforms complex cellular metabolism data into publishable results.

Isolated skeletal myoblasts were cultured on Matrigel® coated 24-well Seahorse XF24 plates at a density of 40,000 cells/well and differentiated into myocyte as described earlier. Experiments were performed on days 4 and 8 of differentiation in vitro.

For fatty acid oxidation analysis, the day prior to the assay growth medium was exchanged for substrate-limited medium (DMEM without glucose, glutamine, sodium pyruvate or bicarbonate (Agilent, Cat. no. 103575-100) supplemented with 0.5 mM glucose, 1 mM glutamate, 0.5 mM carnitine and 1% FBS). On the day of assay sensor calibration was performed according to the manufacturer's instructions and substrate limited medium was exchanged for fatty acid oxidation buffer (111 mM NaCl, 4.7 mM KCL, 1.25 mM CaCl$_2$, 2 mM MgSO$_4$, 1.2 mM NaH$_2$PO$_4$, 5 mM HEPES, 2.5 mM glucose and 0.5 mM carnitine) and the plate was equilibrated at 37 °C for 1 h before the measurements. Cell mitochondrial stress test was performed after adding BSA control or BSA-palmitate (200 µM final concentration). To study the effect of Ppar agonists on fatty acid oxidation, the cells were treated with or without ciprofibrate (50 µM), bezafibrate (500 µM), and fenofibrate (25 µM) in the medium 72 h prior to measurement.

For pyruvate/glucose/glutamine oxidation analysis, calibration sensors were prepared according to the manufacturer's instructions. On the day of assay, differentiation medium was exchanged for DMEM base medium (Agilent, Cat. no. 103575-100) supplemented with 1 mM pyruvate, 2 mM glutamine, and 10 mM glucose, and the plates were equilibrated at 37 °C for 1 h before the measurements.

For all assays, mitochondrial function was probed by the sequential addition of oligomycin (1 µM), FCCP (4 µM), rotenone (1 µM) and antimycin A (5 µM); all final concentrations. Three measurements were performed for each condition. All experiments were normalized to total protein as determined by a BCA protein quantitation assay.

**Transmission electron microscopy**. For TEM analysis, ~5 × 5 mm cubes of quadriceps, gastrocnemius, and triceps were fixed with 2.5% glutaraldehyde diluted with 0.1 M sodium cacodylate buffer and postfixed with 1% osmium tetroxide in Millonig's buffer solution for 2 h, dehydrated, and embedded in epon araldite resin. Ultrathin sections were stained with 2% uranyl acetate for 30 min and lead citrate for 4 min and viewed with a JEOL JEM 1230 Transmission Electron Microscope at 80 kV. Images were captured using a Hamamatsu ORCA-HR digital camera. Three mice per genotype for each timepoint were evaluated. The mitochondrial content was determined from the images at 10,000× magnification using Image J software, (Fiji), (ImageJ 2.0.0-rc-69/1.52n), and calculated as mitochondria count/field by blinded investigators. Point counting was used to estimate mitochondrial volume density and mitochondrial cristae density based on standard stereological methods[45,46]. Only mitochondria profiles of acceptable quality defined as clear visibility and no or few missing spots of the inner membrane were included. Using ImageJ software, a point grid was digitally layered over the micrographic images at 20,000× or 40,000× magnification for mitochondrial volume density and cristae density calculations respectively. Grid sizes of 85 nm × 85 nm and 165 nm × 165 nm were used to estimate mitochondria volume and cristae surface area, respectively. Mitochondria volume density was calculated by dividing the points assigned to mitochondria to the total number of points counted inside the muscle. The mitochondrial cristae surface area per mitochondrial volume (mitochondrial cristae density) was estimated by the formula: mitochondrial cristae density = (4/π) BA, where BA is the boundary length density estimated by counting intersections on test lines multiplied by π/2. In brief, we counted the intersections I(imi) between the inner mitochondrial membrane trace and the test lines and measured the total length of the test line within the mitochondria profile to calculate mitochondrial cristae density = 2. I(imi)/L(mi).

**Western blot analysis (WB) and quantification**. The muscle tissue (~100 mg) was homogenized with a steel bead in 1 ml of cold RIPA buffer containing 1X Proteinase Inhibitor Mix (complete™ Protease Inhibitor Cocktail, Roche, Cat. no.11 697 498 001), 1X PhosStop (Roche, Mannheim Germany, Cat. no.04 906 845 001) using a TissueLyser II instrument (Qiagen) set at 30 strokes/s for 2–4 min. Based on protein quantification results, all samples were adjusted to the final concentration of 2 µg/ul and heat-denatured for 5 min at 99 °C in 2X Laemmli buffer. Proteins were separated by SDS-PAGE and transferred to nitrocellulose membranes. The membranes were incubated in Odyssey blocking solution for 1 h. Total proteins were detected by probing the membranes with appropriate primary antibodies overnight at 4 °C. The following antibodies were used: Chkα (1:1000, Abcam Cat#ab88053), Pparα (1:1000, Abcam, Cat#Ab24509), Pparb (1:1000, Biorad, Cat#AHP1272), Cpt1b (1:1000, Proteintech®, Cat#22170-1-AP), Chkβ

(1:250, Santa Cruz, Cat#398957), GAPDH (1:1000, Cell signaling, Cat#2118), Pparγ (1:500, Santa Cruz, Cat# sc-7273). Proteins were visualized with goat anti-rabbit IRDye-800- or −680-secondary antibodies (1:20,000, LI-COR Biosciences, Cat#926-32211 and Cat#926-68071) or anti-mouse m-IgGκ BP-CFL 790 (Santa Cruz, Cat. no.sc-516181) using an Odyssey imaging system and band density were evaluated using FIJI (NIH).

*Dual-energy X-ray Absorptiometry (DEXA) Analyses*. Post-mortem full-body scans with the head excluded of three wild-type, four knockout, and four heterozygous mice were conducted using dual-energy x-ray absorptiometry (Lunar PIXImus2, GE Medical Systems). Mice were placed in the prostrate position for the scans which were all conducted by the same individual. A quality control scan to calibrate the DEXA device was completed prior to each use.

*Muscle histology characterization using H&E, anti-laminin, and picro-sirius red staining*. Skeletal muscles samples were frozen in liquid nitrogen-cooled isopentane and stored at −80 °C. Transverse cryosections (10-µm thickness) were prepared from frozen muscles and processed for H&E. For anti-laminin staining, after thawing, sections were rehydrated in PBS at room temperature, blocked with 5% HS in 1xPBS for 30 min at room temperature, and incubated in the anti-laminin antibody (Abcam, Cat# ab11575, 1:1000) for 30 min at room temperature. Primary antibody incubation was followed by labeling with the secondary antibody Alexa Fluor® 594 Goat Anti- Rabbit IgG (Cedarlane, Cat# 111-585-006, 1:250) in 5% HS + 1xPBS for 15 min at room temperature. Sections were mounted in fluor-oshield with DAPI (Sigma-Aldrich, #F6057) and were observed under a confocal microscope (Leica TCS SP8 with LIGHTNING). For picrosirius red staining, sections were air-dried completely before being fixed in 4% formaldehyde for 10 min. Fixed slides were transferred to 100% ethanol for 5 min, air-dried for 30 min, and washed in dH$_2$0 for 3 min. Sections were then incubated in a picro-sirius red solution (Sirus red: Sigma-Aldrich, #365548-25G in 1.3% picric acid: Sigma-Aldrich, #P6744-1GA) for 45 min, then washed twice in 0.5% (v/v) acetic acid (Sigma) in water. Sections were then washed in water and dehydrated in a series of ethanol steps before clearing in Xylene for a total of 10 min. Coverslips were then mounted using Permount (Fisher Scientific, Sp-15-500).

**Cell culture metabolite and lipid extraction**. Cells grown on Matrigel® were collected in two washing steps with cold PBS and the Matrigel was removed by centrifugation[47]. Frozen cell pellets were disrupted using probe sonication (2 × 3 s cycles, power setting 1, Fisher Sonic Dismem, model 100). Protein concentration was determined using BCA assay and 100 µg of protein was transferred to a 1.5 ml microcentrifuge tube and the volume was adjusted to 20 µl using water. 5 µl of diluted ILIS standard (a 50% dilution, in methanol, of isotopically labeled amino acids, CIL MSK-A2-S) was added to each sample. Targeted metabolomics analysis was performed using high performance liquid chromatography coupled to a linear ion trap triple-quadrupole tandem mass spectrometer (LC-MS/MS) as previously described[48]. Peak integration was performed using Skyline, an open-source multiple reaction monitoring (MRM) analysis software (version 21.0). An in-house script written in R (version 4.0.2) was used for data QC analysis, normalization, and plotting. Briefly, for each metabolite, the peak intensities of samples were normalized to the mean of the corresponding intensities of the two nearest flanking QC pool samples (which consisted of a mix of all samples included in the analysis) to reduce drift. For lipidomics analysis all samples were adjusted to a final concentration of 120 µg/ml and spiked with 10 µl of internal standard (Avanti Polar Lipids Inc; Catalog Number-330707). 500 µl of chloroform was added to each sample, vortexed for 30 min, and centrifuged to separate phases (5 min at 6000 rpm). The bottom organic phase was transferred into a new Eppendorf and dried under a nitrogen stream. Samples were stored at −80 °C until ready for analysis.

**Quantification and statistical analysis**. All experiments were repeated 3 or more times. Data are presented as mean ± SEM or mean ± SD, as appropriate. For comparison of two groups the two-tailed Student's *t*-test was used unless otherwise specified. Comparison of more than two groups was done by one-way ANOVA followed by the Tukey's Multiple Comparison test. P values < 0.05 were considered significant.

**Reporting summary**. Further information on research design is available in the Nature Research Reporting Summary linked to this article.

## Data availability
Source data that support findings of this study accompanies the manuscript. Any additional source data are available from the corresponding author upon reasonable request. Source data are provided with this paper.

## Code availability
The custom code used in the study for lipidomics analysis is available via Zenodo https://doi.org/10.5281/zenodo.5914355[49]. Download all files into same folder and run analysis.R to reproduce lipidomic analysis plots and tables from this study.

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

## Acknowledgements

We acknowledge funding support from the Canadian Institutes for Health Research (to CRM) and the Atlantic Innovation Fund (to CRM and EH). We thank Gregory Cox for sharing *Chkb* mice.

## Author contributions

M.T., K.N., E.P.H., C.S., G.S.R., and C.R.M. conceived the study. M.T., S.L., S.S., R.N., S.A.R., A.L., M.O.O., K.U., J.R., S.S., M.P., A.M., M.S.M., A.A.M., MaM., J.P.F.M., and MeM. performed experiments. M.T. and C.R.M. wrote the paper.

## Competing interests

The authors declare no competing interests.
