## [Peer Review File · Nature Communications]

Mechanism of action and therapeutic route for a muscular dystrophy caused by a genetic defect in lipid metabolismREVIEWER COMMENTS

Reviewer #1 (Remarks to the Author):

It has been long known that CKB, the first enzyme in PC synthesis, knockout mice develop muscular dystrophy (MD). Initial reports by the Cox/Vance labs in 2006 (Sher et al JBC) showed CKB-/- develop hindlimb MD very early after birth. The authors further observed multiple sites of sarcolemma disruption and the presence of enlarged mitochondria. Cox/Vance did report a small, yet significant reduction in PC levels in hindlimb, but not forelimb. In 2010, the Vance lab (Wu et al) reported further studies into PC metabolism in the hindlimb of CKB-/- mice. While PC levels were reduced, this was surprisingly not due to a reduction in PC synthesis (in fact it was increased). PC turnover was also increased (either from de novo synthesis or VLDL-uptake) in CKB-/- mice due to increased PC-PLC activity. Wu et al again reported that CKB knockout mice have enlarged mitochondria with a reduction in inner membrane potential. CKB-/- hindlimb muscle, but not forelimb, have impaired recovery from chemical injury. The reason for the difference is the higher relative expression of CKa in forelimb muscles.

Tavosoli et al provides a comprehensive mechanistic evaluation of the development of muscular dystrophy in choline kinase b (CKB) knockout mice that builds on previous work. Figures 1 and 2 confirm the CKB-/- mice develop hindlimb muscular dystrophy. It should be pointed out that CKB-/- also have forelimb bone deformity (Sher JBC, Li et al BBA2014) that could impact exercise ability. Interestingly, Tavosoli et al. did not observe a reduction PC, but a large increase in acylcarnitine and cardiolipin. These changes are novel and consistent with alteration in mitochondria morphology. The impairment in mitochondria oxidation eventually results in accumulation of lipid droplets. Tavosoli et al. clearly shows an impairment in PPAR activation in CKB-/- and provides strong evidence that PPAR agonists can improve mitochondria oxidation in isolated myocytes. Overall, the data provided in this manuscript provides a clear mechanistic explanation for altered energy metabolism in an established model of muscular dystrophy.

Specific comments:

1. The authors reproduced most of the data previous reported on the CKB-/- mice except they did not observe any reduction in PC levels in the hindlimb. As the Vance lab published several papers on this topic, the authors should address the difference. For example, how sensitive is summing all individual lipid species to calculate final lipid class? As presented, there appears to be substantial variation in the lipid levels between samples (For example, Figure 3: KO-PC ranges from 106 to 1010).
2. The experiments detailing impaired PPARa signalling is excellent. The in vitro data showing PPAR agonist regulates Chka and potentially improved muscle function is exciting. Clearly the next step is to provide evidence that this could actually work in vivo. It is likely that treatment would have to start immediately after birth since muscle damage occurs very quickly in these mice.
3. The authors state (Page 19) that "Ppar agonists also increased Chka expression in Chkb-/- affected muscle which would enable PC synthesis to now occur in these cells." This statement could have been tested in vitro. It certainly appears from the current work and published reports that flux through the Kennedy pathway and overall PC homeostasis does play a role in the development of MD in the CKB-/- mice. Furthermore, lipid analysis from the in vitro studies would have strengthen the relationship between changes in lipid metabolism with mitochondria function.

Reviewer #2 (Remarks to the Author):

This is an interesting investigation of the muscle disease in mice with Chkb KO. The authors show convincingly that the deficiency is not due to a reduction in phosphatidylcholine, but rather to other defects in lipid metabolism leading to mitochondrial defects This unexpected finding will make an important contribution to the literature. It is further interesting that the authors discover that PPAR agonists ameliorate the disease, in part by upregulation of the homologue Chka.

The muscle physiology studies are well done.

The only significant deficiency is that formally, the authors have not shown a dystrophy, rather they have only shown muscle weakness. There are some basic characterizations that the authors should do to both establish the dystrophy and to better characterize the muscle disease.

1. They should provide muscle weights across a variety of affected (and less- or unaffected) muscles to evaluate the presence of atrophy.
2. They should test for serum creatine kinase, a simple test that measures the presence of systemic muscle fiber damage.
3. They should perform histology on cross sections of TA or other affected muscles, with stains that allow characterization of a) fiber size distribution, b) central nucleation, c) infiltration of mononuclear inflammatory cells, d) presence of fibrosis. This could be as simple as H&E staining and collagen staining (Sirius red or an antibody).

A muscular dystrophy will have an abnormally wide distribution of fiber sizes, abnormal shape (due to expanded endomysium and excess ECM, i.e., fibrosis), evidence of regeneration due to fiber death (centronucleation), and typically an excess of infiltrating cells. Whatever the phenotype is, it will be important to characterize it histologically.

Reviewer #3 (Remarks to the Author):

The manuscript presents novel and scientifically interesting information on the time dependent alterations of metabolic pathways associated with inactivation of the CHKB gene. The authors show, that loss of function is not a change in PC level, but instead leads to a reduced capacity to utilize fatty acids for oxygen production, increased triacylglycerol formation and lipid droplet accumulation. The authors also propose and test a combination of choline supplementation and Ppar-agonist activation as a potential therapeutic approach to treat CHKB-deficiency mediated congenital muscular dystrophy.

The experimental design, the number of experiments/animals, the applied methodology and statistics are all state of the art. The manuscript is clearly structured and well written, despite there are a few language problems and typing errors. The Introduction and Discussion include the most relevant actual literature. The quality of the data and supplementary information is high. The Discussion focuses on the results generated. The data interpretation and the conclusions are supported by the results presented. The significance of the conclusions for the field of lipidomic research and related fields is high. The manuscript is acceptable for publication as it is, after the few language and typing error corrections.

Reviewer #4 (Remarks to the Author):

This is an interesting paper, the work support the conclusions and the paper is well written.

1. The change in PC level is not the major metabolic driver behind this inherited muscular dystrophy in Chkb^{-/-} mice. 2, the mechanistic model for the disease seems to be two temporal stages beginning with an inability to use fatty acids for mitochondrial β -oxidation and a compensatory shunting of fatty acids for storage as TG rich lipid droplets. 3, deficient expression of Ppara, and Ppar β /d, along with their target genes is the main mechanism for the inability of fatty acids to be used for mitochondrial β -oxidation and their shunting into lipid droplets, and finally, 4 the use of Ppar activation along with the metabolic precursor for PC synthesis, Choline, as a potential therapy for this muscular dystrophy.

I have some questions related to these conclusions:

1. In the present study the authors use mouse and cell models to investigate the temporal changes in lipid metabolism in the absence of Chkb gene. Moreover, they tested muscle function in wild type, heterozygous mice and Chkb^{-/-} mice from 5 week to 20 /60 weeks of age. The Chkb^{-/-} mice had lower forelimb strength than wild type mice. The mice in absence of Chkb gene showed a basal level of total distance run was 50 % that of wild type. Finally, the Chkb^{-/-} mice weighted significantly less than the wild type counterparts at all time points . What is the reason for this ? It is not discussed.
2. To understand the temporal development of morphology changes in mitochondria in hindlimb

muscle of Chkb^{-/-} mice, TEM method was used. At 12 days of age, the size of mitochondria increased while the number of mitochondria remained the same. This was accompanied with a 15-fold increase in acylcarnitine level in hindlimb muscle and a 10-fold increase in the level of cardiolipin. Could the increase in acylcarnitines be detected in plasma? The increase in acylcarnitines - is that related to short-, medium and long-chain acylcarnitines. Could these acylcarnitines be biomarkers for defect mitochondria in oxidizing different chain lengths of fatty acids. How was the carnitine level changed?

3. In sharp contrast to 12-day old mice, the acylcarnitine levels were no longer increased in the hindlimb of 30-day old Chkb^{-/-} mice. This was, however, accompanied with a 12-fold increase in the neutral store lipid TG, and at 60 days of age, the number of mitochondria and cristae density decreased significantly, while the size did not change. This implies that the affected muscles with enlarged mitochondria are defective in using fatty acids for the production of cellular energy by mitochondrial β -oxidation. As Chkb^{-/-} muscular dystrophy progresses, the affected muscles appear to adapt to this inability to consume fatty acids by transitioning toward energy storage indicated by the large increase in TG. The lipid droplets were associated with the mitochondria and I wonder how the gene expressions of *dagt2* and *dgat1* were affected. Was de novo lipogenesis and TG biosynthesis affected?

Response to Reviewers

Reviewer #1 (Remarks to the Author)

It has been long known that CKB, the first enzyme in PC synthesis, knockout mice develop muscular dystrophy (MD). Initial reports by the Cox/Vance labs in 2006 (Sher et al JBC) showed CKB^{-/-} develop hindlimb MD very early after birth. The authors further observed multiple sites of sarcolemma disruption and the presence of enlarged mitochondria. Cox/Vance did report a small, yet significant reduction in PC levels in hindlimb, but not forelimb. In 2010, the Vance lab (Wu et al) reported further studies into PC metabolism in the hindlimb of CKB^{-/-} mice. While PC levels were reduced, this was surprisingly not due to a reduction in PC synthesis (in fact it was increased). PC turnover was also increased (either from de novo synthesis or VLDL-uptake) in CKB^{-/-} mice due to increased PC-PLC activity. Wu et al again reported that CKB knockout mice have enlarged mitochondria with a reduction in inner membrane potential. CKB^{-/-} hindlimb muscle, but not forelimb, have impaired recovery from chemical injury. The reason for the difference is the higher relative expression of CKa in forelimb muscles.

Tavosoli et al provides a comprehensive mechanistic evaluation of the development of muscular dystrophy in choline kinase b (CKB) knockout mice that builds on previous work. Figures 1 and 2 confirm the CKB^{-/-} mice develop hindlimb muscular dystrophy. It should be pointed out that CKB^{-/-} also have forelimb bone deformity (Sher JBC, Li et al BBA2014) that could impact exercise ability. Interestingly, Tavosoli et al. did not observe a reduction PC, but a large increase in acylcarnitine and cardiolipin. These changes are novel and consistent with alteration in mitochondria morphology. The impairment in mitochondria oxidation eventually results in accumulation of lipid droplets. Tavosoli et al. clearly shows an impairment in PPAR activation in CKB^{-/-} and provides strong evidence that PPAR agonists can improve mitochondria oxidation in isolated myocytes. Overall, the data provided in this manuscript provides a clear mechanistic explanation for altered energy metabolism in an established model of muscular dystrophy.

We thank Reviewer 1 for their careful consideration of our manuscript. We tried to allay each concern of Reviewer 1 in our response below and changes to the manuscript.

We agree that the *Chkb*^{-/-} forelimb bone deformity could impact exercise ability. We added language to highlight this point.

Specific comments

1. The authors reproduced most of the data previous reported on the CHKB^{-/-} mice except they did not observe any reduction in PC levels in the hindlimb. As the Vance lab published several papers on this topic, the authors should address the difference. For example, how sensitive is summing all individual lipid species to calculate final lipid class? As presented, there appears to be substantial variation in the lipid levels between samples (For example, Figure 3: KO-PC ranges from 106 to 1010).

In our study we used a different method for lipid analysis (High resolution tandem mass spectrometry and lipidomics analysis) compared to the earlier studies by the Vance lab (the high-performance liquid chromatography method of Bergo et al. with minor modifications). Our data is consistent with first report from Vance's lab showing there was no statistical difference in total PC mass in the *Chkb*^{-/-} hindlimb, liver, brain, kidney, and heart (Sher, Aoyama et al. (2006) J Biol Chem 281(8): 4938-4948.) **This Vance study is now discussed/referenced on p11 of the Results.**

Each discrete point on the Figure 3 graph represents the average of a unique PC species (fatty acyl composition) from mice of that genotype. **We have added a statement to note this in the legend of Figure 3.** We did not perform a comparison of the three genotypes of the number of *individually* observed lipid species for PC. Rather, all statistical comparisons were performed pair-wise and log scaled. That is, a fold-change was calculated between the knockouts and controls for each unique lipid species and a non-parametric statistical test was used to assess if the fold-changes were statistically significant from 1 in aggregate. Lipid concentrations extracted from the LipidSearch software were analyzed with a stringent in-house script using the R programming language. The data was filtered to exclude any peak concentration estimates with a signal to noise ratio (SNR parameter) of less than 2.0 or a peak quality score (PQ parameter) of less than 0.8. The individual concentrations were then gathered together by lipid identity and grouped within the broader categories of PC. There were never any statistical differences in PC mass or species between groups.

For lipid quantification, our MS approach used the SPLASH® Lipidomix® Mass Spec Standard as internal standards which includes deuterium labeled forms of all the major lipid classes including PC, which is considered more robust than using a single lipid species as an internal standard. **This information has been added to the Methods section.**

2. The experiments detailing impaired PPARα signalling is excellent. The in vitro data showing PPAR agonist regulates Chka and potentially improved muscle function is exciting. Clearly the next step is to provide evidence that this could actually work in vivo. It is likely that treatment would have to start immediately after birth since muscle damage occurs very quickly in these mice.

To address the potential for Ppar agonists +/- choline to correct the defective *Chkb*^{-/-} phenotypes we performed substantive further experimentation. This included lipidomics analysis of differentiated myoblasts from wild type and *Chkb*^{-/-} mice treated Ppar agonists +/- choline. We observed that both the increase in AcCa, as well as that of TG and DG, were normalized (**new data found in Fig. 7A,B and Results page 20**). These extra experiments add substantive mechanism to how Ppar agonists are reversing the metabolic and injury defects observed in isolated muscle from *Chkb*^{-/-} mice. We do agree with the importance of studying the effects of Ppar activation plus choline supplementation in *Chkb*^{-/-} mice. We plan to pursue the preclinical development of novel therapeutics targeting CHKB associated musculoskeletal disorder, however preclinical evaluation of Ppar activation plus choline supplementation in *Chkb*^{-/-} mice requires a ~2 year comprehensive analysis; we feel it is outside of the scope of this manuscript.

3. The authors state (Page 19) that “Ppar agonists also increased Chka expression in *Chkb*^{-/-} affected muscle which would enable PC synthesis to now occur in these cells.” This statement could have been tested in vitro. It certainly appears from the current work and published reports

that flux through the Kennedy pathway and overall PC homeostasis does play a role in the development of MD in the *CKB*^{-/-} mice. Furthermore, lipid analysis from the in vitro studies would have strengthened the relationship between changes in lipid metabolism with mitochondria function.

We used targeted metabolomics of choline and phosphocholine to determine if the addition of Ppar agonists improved flux through the choline kinase step based on the changes in mass of choline and phosphocholine of differentiated myoblasts from *Chkb*^{-/-} mice (as would be predicted based on the increase in *Chka* expression due to PPAR addition). This was found to be the case (**new data found in Fig 6. G,H,I,J,K and Results page 19**); we also reference/discuss a previous Vance paper (Wu et al (2009) *BBA 1791:347-356*) that monitored [³H]choline into hindlimb muscle of *Chkb*^{-/-} mice and noted that there was a defect in PC synthesis and an increase in the level of choline and decrease in phosphocholine.

Reviewer #2 (Remarks to the Author)

This is an interesting investigation of the muscle disease in mice with *Chkb* KO. The authors show convincingly that the deficiency is not due to a reduction in phosphatidylcholine, but rather to other defects in lipid metabolism leading to mitochondrial defects. This unexpected finding will make an important contribution to the literature. It is further interesting that the authors discover that PPAR agonists ameliorate the disease, in part by upregulation of the homologue *Chka*.

The muscle physiology studies are well done. The only significant deficiency is that formally, the authors have not shown a dystrophy, rather they have only shown muscle weakness. There are some basic characterizations that the authors should do to both establish the dystrophy and to better characterize the muscle disease.

We thank Reviewer 2 for their careful consideration of our manuscript. We tried to allay each concern of Reviewer 2 in our response below and changes to the manuscript.

1. They should provide muscle weights across a variety of affected (and less- or unaffected) muscles to evaluate the presence of atrophy.

We agree with the Reviewer regarding the importance of characterizing the muscular dystrophy in *Chkb*^{-/-} mice. We have now added data analyzing muscle weights across a variety of affected (EDL, Gastroc, Quad and TA) and unaffected muscles (Triceps) to evaluate the presence of atrophy. This data is now reported (**new data found in Fig 1G and Results page 8**).

2. They should test for serum creatine kinase, a simple test that measures the presence of systemic muscle fiber damage.

We have now added data for serum creatine kinase levels at 3 time points during the 20 weeks phenotyping. (**new data found in Fig 1F and Results page 8**). The increase in CK levels in *Chkb*^{-/-} mice is mild compared to dystrophic mice with compromised sarcolemmal membrane integrity, where levels of plasma CK are significantly increased (20-fold). Our data is consistent with other studies (Sher, Aoyama et al. 2006) and highlight the fact that sarcolemmal membrane integrity is only mildly affected in *Chkb*^{-/-} mice.

3. They should perform histology on cross sections of TA or other affected muscles, with stains that allow characterization of a) fiber size distribution, b) central nucleation, c) infiltration of mononuclear inflammatory cells, d) presence of fibrosis. This could be as simple as H&E staining and collagen staining (Sirius red or an antibody). A muscular dystrophy will have an abnormally wide distribution of fiber sizes, abnormal shape (due to expanded endomysium and excess ECM, i.e., fibrosis), evidence of regeneration due to fiber death (centronucleation), and typically an excess of infiltrating cells. Whatever the phenotype is, it will be important to characterize it histologically.

New histology includes picosirius staining, confocal microscopy of laminin, and H&E staining to further evaluate the muscular dystrophy phenotypes (**new data found in Suppl Fig. 2 A,B,C and Results pages 8-9**). Muscular dystrophy phenotypes were observed.

Reviewer #3 (Remarks to the Author)

The manuscript presents novel and scientifically interesting information on the time dependent alterations of metabolic pathways associated with inactivation of the CHKB gene. The authors show, that loss of function is not a change in PC level, but instead leads to a reduced capacity to utilize fatty acids for oxygen production, increased triacylglycerol formation and lipid droplet accumulation. The authors also propose and test a combination of choline supplementation and Ppar-agonist activation as a potential therapeutic approach to treat CHKB-deficiency mediated congenital muscular dystrophy. The experimental design, the number of experiments/animals, the applied methodology and statistics are all state of the art. The manuscript is clearly structured and well written, despite there are a few language problems and typing errors. The Introduction and Discussion include the most relevant actual literature. The quality of the data and supplementary information is high. The Discussion focuses on the results generated. The data interpretation and the conclusions are supported by the results presented. The significance of the conclusions for the field of lipidomic research and related fields is high. The manuscript is acceptable for publication as it is, after the few language and typing error corrections.

We thank Reviewer 3 for their careful consideration of our manuscript and their positive comments.

Reviewer #4 (Remarks to the Author)

This is an interesting paper, the work support the conclusions and the paper is well written.

1. The change in PC level is not the major metabolic driver behind this inherited muscular dystrophy in Chkb^{-/-} mice. 2, the mechanistic model for the disease seems to be two temporal stages beginning with an inability to use fatty acids for mitochondrial β -oxidation and a compensatory shunting of fatty acids for storage as TG rich lipid droplets. 3, deficient expression of Ppara, and Pparb/d, along with their target genes is the main mechanism for the inability of fatty acids to be used for mitochondrial β -oxidation and their shunting into lipid droplets, and finally, 4 the use of Ppar activation along with the metabolic precursor for PC synthesis, choline, as a potential therapy for this muscular dystrophy.

I have some questions related to these conclusions:

1. In the present study the authors use mouse and cell models to investigate the temporal changes in lipid metabolism in the absence of *Chkb* gene. Moreover, they tested muscle function in wild type, heterozygous mice and *Chkb*^{-/-} mice from 5 week to 20 /60 weeks of age. The *Chkb*^{-/-} mice had lower forelimb strength than wild type mice. The mice in absence of *Chkb* gene showed a basal level of total distance run was 50 % that of wild type. Finally, the *Chkb*^{-/-} mice weighted significantly less than the wild type counterparts at all time points. What is the reason for this ? It is not discussed.

We thank Reviewer 4 for their careful consideration of our manuscript. We agree with reviewer's comment regarding the importance of body weight differences among genotypes. We have now added a supplementary figure to measure tissue composition (fat mass & lean mass) and bone mineral density in 14 weeks old mice by Dual-Energy X-ray Absorptiometry (DEXA) (**new data found in Suppl. Fig. 1A,B,C,D and results page 6**). We also added language that significant decrease in *Chkb*^{-/-} body weight and body composition highlights a potential role for *Chkb* in regulating growth and metabolism.

2. To understand the temporal development of morphology changes in mitochondria in hindlimb muscle of *Chkb*^{-/-} mice , TEM method was used. At 12 days of age , the size of mitochondria increased while the number of mitochondria remained the same. This was accompanied with a 15-fold increase in acylcarnitine level in hindlimb muscle and a 10-fold increase in the level of cardiolipin. Could the increase in acylcarnitines be detected in plasma? The increase in acylcarnitines - is that related to short-,medium and long-chain acylcarnitines. Could these acylcarnitines be biomarkers for defect mitochondria in oxidizing different chain lengths of fatty acids. How was the carnitine level changed? Short-chain fatty acids (five or fewer carbons), Medium-chain fatty acids (6 to 12 carbons), Long-chain fatty acids (13 to 21 carbons) and Very long chain fatty acids (22 or more carbons).

Our data shows that the increase in AcCa levels is temporal and recedes as the disease progresses. Based on the transient increase in AcCa levels in localized (affected) muscles, plasma AcCa levels might not be a reliable way to be used as early lab markers for this disease. More importantly, other studies reported normal plasma levels of AcCa in *CHKB*^{-/-} patients (Quinlivan, Mitsuhashi et al. 2013, Chan, Ho et al. 2020, Kutluk, Kadem et al. 2020). Further, when determining AcCa levels in affected muscle, our analysis detected a significant increase in all AcCa fatty acid chain lengths including medium-chain fatty acids, long-chain fatty acids and very long chain fatty acids in both 12 day old hindlimb and differentiated myocytes in culture (precise data could be provided). We did not detect any short chain fatty acid containing AcCa in mouse muscle. There was no specific type of fatty acid change apparent for AcCa across any genotype or treatment. **We state this in the Results on page 11.**

3. In sharp contrast to 12 -day old mice, the acylcarnitine levels were no longer increased in the hindlimb of 30-day old *Chkb*^{-/-} mice. This was, however, accompanied with a 12-fold increase in the neutral store lipid TG, and at 60 days of age, the number of mitochondria and cristae density decreased significantly, while the size did not change. This implies that the affected muscles with enlarge mitochondria are defective in using fatty acids for the production of cellular energy by mitochondrial β -oxidation. As *Chkb*^{-/-} muscular dystrophy progresses, the affected muscles appear to adapt to this inability to consume fatty acids by transitioning toward energy storage indicated by the large increase in TG. The lipid droplets were associated with the

mitochondria and I wonder how the gene expressions of *dgat2* and *dgat1* were affected. Was de novo lipogenesis and TG biosynthesis affected?

We have determined the gene expressions of *dgat2* and *dgat1* (**new data found in Suppl. Fig. 3E and Results page 13**). Consistent with increased lipid droplets in hindlimb muscles from *Chkb*^{-/-} mice, the expression levels of the triglyceride synthesis enzymes *Dgat1* and *Dgat2* were increased 3.5 and 3.3 fold in *Chkb*^{-/-} hindlimb muscles compared to the wild type. This is also stated in the Results section.

REVIEWERS' COMMENTS

Reviewer #1 (Remarks to the Author):

The authors have addressed all of my concerns. The paper has been significantly improved.

Reviewer #2 (Remarks to the Author):

The authors have been responsive to the suggestion to include more data on the muscular dystrophy phenotype as well as characterizing various muscles, affected and non-affected. The study is much improved.

Response to reviewers' comments

The reviewers required no additional modifications to the manuscript. We would like to thank the reviewers for their help and time.